# Pathologies of Precursor Lesions of Biliary Tract Carcinoma

**DOI:** 10.3390/cancers14215358

**Published:** 2022-10-30

**Authors:** Yasuni Nakanuma, Yuko Kakuda, Takashi Sugino, Yasunori Sato, Yuki Fukumura

**Affiliations:** 1Department of Diagnostic Pathology, Shizuoka Cancer Center, Shizuoka 411-8777, Japan; 2Department of Diagnostic Pathology, Fukui Prefecture Saiseikai Hospital, Fukui 918-8503, Japan; 3Department of Human Pathology, Kanazawa University Graduate School of Medicine, Kanazawa 920-8641, Japan; 4Department of Human Pathology, Juntendo University School of Medicine, Tokyo 113-8421, Japan

**Keywords:** bile duct, gallbladder, precursor lesion, high-grade biliary intraepithelial neoplasm, intraductal papillary neoplasm, intracholecystic papillary neoplasm, biliary tract carcinoma, pancreatobiliary system

## Abstract

**Simple Summary:**

Carcinomas and precursor lesions of the biliary tract belong to a spectrum of pancreatobiliary neoplasms that share common histology and cell lineages. High-grade biliary intraepithelial neoplasm (high-grade BilIN), intraductal papillary neoplasm of bile duct (IPNB), and intracholecystic papillary neoplasm of the gallbladder (ICPN) are the main biliary tract precursors. High-grade BilINs are microscopically identifiable intraepithelial neoplastic lesions, whereas IPNB and ICPN are grossly visible intraductal or intraluminal preinvasive neoplasms in the bile duct and gallbladder, respectively. These neoplasms show characteristic histologic features according to the four subtypes based on cell lineages and two-tiered grading, and they are not infrequently associated with foci of stromal invasion at the time of surgical resection. In addition, these precursors, particularly high-grade BilINs, are frequently identified in the biliary mucosa surrounding biliary tract carcinomas (BTCs). Taken together, the progression of these precursors to invasive carcinoma may be a major process in biliary carcinogenesis.

**Abstract:**

Carcinomas and precursor lesions of the biliary tract belong to a spectrum of pancreatobiliary neoplasms that share common histology and cell lineages. Over the past two decades, preinvasive precursors to biliary tract carcinomas (BTCs) have been identified such as high-grade biliary intraepithelial neoplasm (high-grade BilIN), intraductal papillary neoplasm of bile duct (IPNB) and intracholecystic papillary neoplasm of the gallbladder (ICPN). While a majority of these precursors may arise from the biliary tract mucosa, some originate from the peribiliary glands and Rokitansky-Aschoff sinuses in the walls of the biliary tract. High-grade BilIN is a microscopically identifiable intraepithelial neoplasm of the biliary tract, whereas IPNB and ICPN are grossly visible intraductal or intraluminal preinvasive neoplasms in the bile duct and gallbladder, respectively. These neoplasms show characteristic histologic features according to four cell lineages and two-tiered grading, and show intraepithelial spreading to the surrounding mucosa and involve non-neoplastic glands in the walls of the biliary tract. These precursors are not infrequently associated with stromal invasion, and high-grade BilIN, in particular, are frequently identified in the surrounding mucosa of BTCs. Taken together, it seems likely that progression from these precursors to invasive carcinoma is a major process in biliary carcinogenesis.

## 1. Introduction

Biliary tract carcinomas (BTCs) are aggressive epithelial malignancies that can arise in the biliary tract [1,2,3]. Over the past two decades, reports have indicated that BTCs are preceded by non-invasive, in situ neoplasms (precursors), including microscopically identifiable atypical biliary epithelial hyperplasia, and grossly visible, papilloma or papillomatosis [4,5,6]. The WHO 2019 Tumor Classification of Digestive System [1] proposed several types of precursor lesions of the biliary tract: (i) microscopically identifiable, biliary intraepithelial neoplasm (BilIN); (ii) grossly visible, intraductal papillary neoplasm of bile duct (IPNB); (iii) grossly visible, intracholecytic neoplasm of the gallbladder (ICPN); (iv) pyloric gland adenoma (PGA) of the gallbladder; and (v) hepatobiliary mucinous cystic neoplasm (MCN). These precursors have been rapidly incorporated into management algorithms for biliary tract tumors, which have presented an increasing incidence over the past 40 years [7,8,9,10,11].

Thus far, the pathologies of precursor lesions of the biliary tract and their progression to invasive carcinomas have been studied in comparison to their counterparts in the pancreatobiliary system, particularly to their pancreatic counterparts such as pancreatic intraepithelial neoplasm (PanIN), intraductal papillary mucinous neoplasms (IPMNs), intraductal oncocytic papillary neoplasm (IOPN), and MCN [1,12,13,14,15]. The precursors of the biliary tract also shares common features with precursor lesions of the ampulla, particularly intestinal adenoma [1]. Adenocarcinomas of the pancreatobiliary system, including BTCs, also share common histological and immunohistochemical features and cell lineages [2,16,17]. In this context, a comprehensive pathologic analysis of precursors and carcinomas of the pancreatobiliary system as a spectrum will be useful for the evaluations of these neoplasms [16,17,18].

There have been many overviews and discussions on BTCs referring to recent progress and consensus reached on clinical and therapeutical aspects and molecular and genetic alterations [2,3,4,10,19,20], although such overviews of the precursors of the biliary tract are very limited [4,7,8,21,22]. Therefore, the pathologies of precursors of the biliary tract must be reviewed to facilitate a better understanding of biliary tract neoplasms, including BTCs.

Herein, we describe BTCs and precursors of the biliary tract that are pancreatobiliary neoplasms. Then, we discuss in detail recent progress on the pathologic features of the precursors of the biliary tract, particularly high-grade BilIN, IPNB and ICPN, referring to BTCs and their pancreatic and ampullary counterparts.

## 2. Biliary Tract Carcinomas and Precursors as Pancreatobiliary Neoplasms

Carcinomas and precursors of the pancreatobiliary system share similar pathological and immunohistochemical features, including four cell lineages [12,18,23,24,25]. The following characteristics of the pancreatobiliary system may explain these similarities.

### 2.1. Backgrounds

#### 2.1.1. Embryology and Anatomy

Embryologically, hepatobiliary and pancreatic systems are derived from the foregut endoderm [2,23,24,25,26]. Intrahepatic small bile ducts arise from hepatic progenitor cells via ductal plates, whereas the extrahepatic biliary tract and ventral pancreas, including the ampulla, arise from pancreato-extrahepatic biliary progenitor cells expressing transcription factors PDX1 and HES1 [12,27,28]. FGF10 signaling from the adjacent mesenchyme may also regulate hepatopancreatic system differentiation toward different organ fates [29]. SOX9 is expressed throughout the biliary and pancreatic duct epithelial cells, which are connected to the intestinal stem cell zone [30]. SOX9 is involved in the interdependence between the structure and homeostasis of the pancreatobiliary system and intestine. The similar histology and shared expression of immunophenotypes between pancreatobiliary carcinoma and precursors may be related to the unique pancreatobiliary anatomy/embryology and the autocrine/paracrine effects and switch-on/off of these signaling factors.

#### 2.1.2. Multiple Cell Types Composing the Individual Regions of the Pancreatobiliary System

Recent studies suggest that different and diverse tumor profiles may be significantly influenced by the existence of multiple cells of origin [2,31]. The individual anatomical regions of the pancreatobiliary system are composed of multiple epithelial cell types. Intrahepatic large and extrahepatic bile ducts and pancreatic ducts are lined by simple columnar epithelia and accompanied by peribiliary and pancreatic duct glands in their walls and periductal tissues, respectively [25,28,32]. Peribiliary glands express pancreatic and endodermal markers, and also contain biliary stem/progenitor cells that generate mature biliary-pancreatic lineage cells, while pancreatic duct glands contain progenitor cells for pancreatic duct epithelial cells [23,33]. The gallbladder mucosa is lined by surface columnar epithelia containing multipotent endodermal stem cells and Rokitansky-Aschoff sinuses (R-A sinuses) are frequently seen in their walls [23,32,34]. In ampullary regions, the duodenum-facing surface is partially covered by intestinal-type epithelia and the ampullary lumen is covered by pancreatobiliary epithelia. Pancreatobilairy type glands are observed on the walls, and they show morphologic characteristics of pancreatobiliary and intestinal differentiation and often express the intestinal transcription factor CDX2 [16]. Pathologically, these regions present various metaplastic changes such as gastric foveolar and pyloric glands and intestinal metaplasia. Differences and similarities in immunohistochemical/histological subtypes among pancreatobiliary carcinomas and their precursors may be related to the neoplastic transformation of multiple cell types in individual anatomical regions [12,32,35,36,37].

#### 2.1.3. Four Cell Lineages

Pancreatobiliary carcinomas and precursors commonly present one or more of the following four subtypes based on cell lineage: intestinal, pancreatobiliary (PB), gastric and oncocytic [38,39,40,41]. Their main features are listed in Table 1. The predominant type among the four cell lineages and their degrees of differentiation vary among carcinomas and precursors in individual anatomical parts.

### 2.2. Biliary Tract Carcinomas (BTCs)

Currently, carcinomas of the pancreatobiliary system are classified into six categories based on the anatomical site of origin (AJCC, UICC-TNM 7th edition); ampullary carcinoma (AmCA), pancreatic duct adenocarcinoma (PDA), and BTCs [42,43]. BTCs are further subdivided into gallbladder carcinoma (GBC) and cholangiocarcinoma (CCA), which is further classified into intrahepatic CCA (iCCA), perihilar CCA (pCCA) and distal CCA (dCCA) according to its main anatomical location [42,43]. pCCA and dCCA are collectively referred to extrahepatic cholangiocarcinoma (eCCA), and iCCA is further divided into small duct type (SD-iCCA) and large duct type (LD-iCCA) [1,10].

#### 2.2.1. Epidemiology

The incidence of GBC varies both geographically and ethnically, its incidence being the highest in Chile, of which females comprise 27.3 cases per 100,000 person-year [1,44]. Its incidence is also higher in Indica, Eastern Asia, and Central and Eastern European countries [1,45]. In Chile, GBC occurs predominantly in females with gallstones, while the association with gallstones in Eastern Asia is weak [1,44,45]. The incidence of extrahepatic CCAs ranges from 0.53 to 2 cases per 100,000 person-year worldwide, with higher frequency in East Asian countries [46]. In some regions of East-Asia, particularly the Republic of Korea and Thailand, liver fluke infestations and hepatolithiasis are endemic and these risks could be related to cholangiocarcinogenesis [46].

#### 2.2.2. Gross and Microscopic Features of BTCs

dCCAs, pCCAs, LD-iCCAs and GBCs are grossly classified into intraluminal polypoid (papillary or villous), nodular, and flat growth types [1,47,48,49]. The polypoid type presents intraluminal growth of carcinoma in the bile duct and gallbladder lumen, while the nodular and flat types typically present nodular or flat thickening of the bile duct and gallbladder wall due to infiltrative growth of carcinoma cells with desmoplastic reaction. In contrast, SD-iCCA usually presents as a mass-forming lesion in the liver [1,10,47].

Microscopically, dCCAs, pCCAs, LD-iCCAs and GBCs mostly present well- to moderately- differentiated duct-like structures lined by cuboidal or columnar adenocarcinomas with micropapillary or solid/cord-like components [1,10,47]. Moreover, dCCAs, pCCAs, LD-iCCAs, and GBCs are mostly, mucin-positive material in the cytoplasm, acinar lumen, or luminal border [1,4].

#### 2.2.3. Four Subtypes of BTCs as a Spectrum of Pancreatobiliary Carcinoma

Pancreatobiliary carcinomas frequently present with the PB subtype (ordinary type) and less frequently present with the intestinal subtype, although a variable admixture of these two subtypes has been immunohistochemically observed [41,48,49,50,51,52]. The former is frequent and has an unfavorable post-operative prognosis compared with the latter [41]. In addition, some cases of pancreatobiliary carcinoma resemble gastric-type adenocarcinoma and rarely oncocytic carcinoma [16,17]. However, iCCA shows diverse histologic and phenotypic profiles that are different from others of BTCs [10,47].

Several immunohistochemical classification schemes have been devised [49,50,51]. For example, Fernadez Moro et al. [16], performed hierarchical clustering and integrative differential analysis of immunohistochemical expression and identified three distinct subtypes based on their predominant features: PB, intestinal, and intrahepatic CA. The intestinal subtype [16,49,50,51] expressed CK20, MUC2, and CDX2. Tumors of the intestinal type co-expressed PB markers to a variable extent. The PB subtype express a considerable number of markers, including cytokeratins (CK7, CK17, and CK19), mucins (MUC1 and MUC5AC), and tumor-associated epithelial markers (CA19-9, monoclonal CEA, CA125, and maspin). The intrahepatic CA subtype expresseses CK7, CK19, BerEP4, and polyclonal CEA. Moreover, they identified cytoplasmic WT1 as a novel marker for iCCA subtype [16]. This immunohistochemical integrative analysis [16] revealed that the PB subtype is the most common subtype in PDAC, dCCA, AmCA and GBC. Half of the pCCA cases were of the PB subtype, however, only approximately one-third of the iCCA cases were of the PB type. In contrast, the intestinal subtype was relatively common in AmCA, infrequent in other carcinomas, and occurred only occasionally in iCCA. The intrahepatic CA subtype was frequent in iCCA and occurred occasionally in pCCA, GBC, pCCA, and AmCA, and rarely in PDAC.

#### 2.2.4. LD-iCCA, pCCA, dCCA, and GBC Share Variable Pathologic and Immunohistochemical Features

Recent studies have shown that LD-iCCA is different from SD-iCCA but more similar to pCCA and dCCA [10,53]. Grossly, SD-iCCA presents a mass-forming growth pattern, whereas LD-iCCA presents with intraluminal polypoid, nodular or flat sclerosing carcinoma growth [1,10,47]. SD-iCCA presents more heterogeneous histological features and unique immunohistochemical features that are different from those of LD-iCCA, pCCA and dCCA [10,52]. The latter three share similar histology and molecular features, frequently show perineural and lymphovascular invasion, and histologically and immunohistochemically resemble PDAC and GBC [27]. In addition, LD-iCCA and pCCA more frequently show loss of SMAD4 expression and MDM2 amplifications, whereas SD-iCCA more frequently shows loss of BAP1 expression and IDH1 mutations [1,10,48,53].

These findings strongly suggest that pancreatobiliary carcinomas share common histologic and phenotypic features in a variable proportion except for SD-iCCA [10,48,53].

#### 2.2.5. Genetic Alterations in LD-iCCA, pCCA, dCCA, and GBC

More than 50% of GBCs and approximately 50% of extrahepatic CCAs harbor alterations in *TP53* alteration [53,54]. Common mutations in GBC include *CDKN2A* or *CDKN2B* (19%), *ARID1A* (13%), *PIK3CA* (10%) and *CTNNB1* (19%), and amplifications of ERBB2 have been also reported [55,56,57]. Specific extrahepatic CCA alterations include *PRKACA/PRKACB* fusion, *ELF3* mutation and *ARID1B* mutation [58]. *KRAS* mutations were observed in 20–30% of extrahepatic CCAs, while an increase in *KRAS* mutations are closely related to GBCs with pancreatobiliary malunion [59].

Genetic alterations in the precursors of BTCs are discussed in individual lesions in the following sections.

### 2.3. Precursors of the Pancreatobiliary System

The concept of epithelial tumors arising from non-invasive intraepithelial dysplasia (precursors) is well-established in various tumors [1,60,61]. The current WHO tumor classification proposes several precursors in individual segments of the pancreatobiliary system that share common but variably different features and cell lineages (Table 2) [1].

#### 2.3.1. Precursors of the Biliary Tract

Many cases of BTCs have been reported to develop through multistep carcinogenesis and are preceded by unique precursors: high-grade BilIN, IPNB, ICPN, MCN and PGA [1,22,61,62,63]. These precursors will be discussed in detail later.

#### 2.3.2. Precursors of the Pancreas and Ampulla

In the pancreas, IPMN, IOPN, intraductal tubulopapillary neoplasm (ITPN), PanIN and MCN are precursors, while in the ampulla, intestinal adenoma and intra-ampullary papillary-tubular neoplasm (IAPN) are precursors [1]. IPNBs and ICPNs may represent the bile duct and gallbladder equivalent to pancreatic IPMN and IOPN, respectively, whereas high-grade BilIN represents the biliary tract equivalent to high-grade PanIN [12,24]. The intestinal subtypes of IPNB and ICPN may be equivalent to intestinal adenomas of the ampulla. MCN is similarly found in the pancreas and hepatobiliary system [1].

### 2.4. Treatments and Diagnostic Approaches of BTCs and Precursors of the Biliary Tract

The treatments of BTC is dependent on its anatomical location and stages based on the TNM classification [42,43]. Biliary tract precursors, particularly IPNBs, are not infrequently associated with invasive carcinoma [7,8,38], thus, they are recommended to be treated similarly to BTCs [7,8]. Recent progress in biomarkers and diagnostic and therapeutic approaches for biliary neoplasms, including BTCs, has been reported in the IASGO textbook [64].

## 3. Common Precursors of the Biliary Tract

High-grade BilIN, IPNB and ICPN are common precursors of the biliary tract, and the majority of BTCs develop through these precursors [1,4,7,18,22]. While IPNB in the bile duct and ICPN in the gallbladder present several common pathological features, they are separately described here to clarify these neoplasms.

### 3.1. BilIN

BilINs are found in the intrahepatic large and extrahepatic bile ducts and the gallbladder [4,21,22,26,62,65,66,67,68,69]. These are microscopic, non-invasive lesions that have been previously named dysplasia or atypical epithelial hyperplasia [6]. BilIN is reportedly the most frequent precursor lesion of flat, nodular, or sclerosing CCA and GBC [1,15,62,70,71]. In 2007, an international interobserver agreement study on the diagnosis of non-invasive biliary neoplastic lesions was conducted to obtain a consensus on the terminology and grading in reference to PanIN in the pancreas [67]. BilINs were classified into BilIN-1, -2, and -3 according to their cytological and structural alterations [66]. The 2019 WHO classification histologically stratified BilIN using a two-tiered classification (high vs. low) (Table 3) [1]. High grade BilIN (high-grade dysplasia, in situ carcinoma) is staged as an in situ carcinoma according to UICC/TNM classification [1,42,43]. BilIN of the gallbladder is occasionally called dysplasia, and in South America and Japan, high-grade BilINs could be alternatively called in situ carcinomas [1,22].

#### 3.1.1. Clinical Features and Background

##### Clinical Features

Typically, BilINs are incidentally detected in the biliary tract mucosa in cases of chronic biliary diseases such as hepatolithiasis, cholelithiasis, primary sclerosing cholangitis (PSC) and pancreatobiliary malunion [4,22,66,69], although they are also infrequently deteced in the bile ducts of non-biliary cirrhotic livers (HCV and alcohol related) as well as familial adenomatous polyposis [66]. While BilINs do not produce clinical symptoms and are not detectable on imaging studies [1], high-grade BilINs of the gallbladder can be detected in cholecystectomized cases with a preoperative suspected diagnosis of malignancy by imaging [72]. The prevalence of high-grade BilINs of the gallbladder with lithiasis reportedly ranges from 0.4% to 3.5% depending on the geographical area [1,18]. High-grade BilINs frequently affect middle-aged and elderly people and are more prevalent in females [1,18,22].

##### Backgrounds

In addition to the above-mentioned backgrounds for BilIN, atypical intraepithelial lesions similar or identical to high-grade BilIN without underlying infiltrative carcinoma are often encountered in the bile duct margins of surgically resected cases of CCA [72,73,74]. Similar lesions without the underlying infiltrative carcinoma are also frequent in the surrounding mucosa adjacent to invasive CCA and GBC [22,71]. The absence of infiltrating carcinoma under these intraepithelial neoplasms suggests that they do not represent surface cancerization of invasive carcinoma [74,75,76,77,78,79,80,81,82,83,84,85,86] but rather may be a remnant or preceding components of high-grade BilINs from which CCA or GBC might have developed. High-grade BilINs at the surgical margin of the bile duct are usually continuous with atypical intraepithelial lesions in the surrounding mucosa adjacent to CCAs, thus corresponding to laterally spreading high-grade BilINs [77,78].

However, whether atypical intraepithelial lesions accompanying or associated with an underlying infiltrative carcinoma in the biliary tract wall can be regarded as high-grade BilINs remains unclear. Frequently direct continuation occurs between these intraepithelial atypical epithelia and the underlying infiltrative carcinoma, which probably reflects cancerization of the bile duct or gallbladder by invasive carcinoma (please see the section “Related diseases and differential diagnosis”) as discussed in PDAC and high-grade PanIN [74,75,76].

##### Epidemiology

Low-grade BIlIN is seen in as many as 15% of gallbladders with lithiasis, and high-grade BilIN in 1–3.5%. in regions where GBC is endemic [78,79]; incidences were reported in <5% and <0.1% in North America [1,81]. The epidemiology of low-grade and high-grade BilIN in the bile ducts has not been reported because they are seldom biopsied. However, high-grade BilIN lesions are frequently found in the surrounding mucosa of BTC [22,80,81], although the epidemiology of such BilINs has not been surveyed epidemiologically.

#### 3.1.2. Gross and Microscopic Features

High-grade BilINs are grossly unnoticeable or present with recognizable granular and rough mucosa [22,71,82]. The area of high-grade BilINs ranges from 0.5 to 6 cm, indicating that the areas showing high-grade BilIN involvement are already considerable, when detected, suggesting an insidious intraepithelial spreading of neoplastic epithelia [22,71,72,82].

Histologically, their basic structures present with flat/pseudopapillary, or micropapillary patterns, and many high-grade BilIINs are associated with few fibrovascular connective tissues (Figure 1A,B) [1,4,82,83]. Micropapillary structures may reflect physiological short foldings or papillary structures inherent to the bile duct and gallbladder mucosa [34,71]. However, micropapillary patterns higher than the surrounding non-neoplastic mucosa, may reflect unique tumor-specific structures. The micropapillary pattern is multiple but less common than the flat pattern [71,80,84]. Furthermore, some high-grade BilINs show complicated lesions and bizarre cellular and nuclear changes, particularly among micropapillary components, thus reflecting a relatively aggressive characteristic [71]. However, even simple, flat, high-grade BilINs without architectural complexity may show progression [22]. The flat type is undetected during macroscopic examination, whereas the micropapillary type may be recognized by mucosal thickening, velvety texture and /or fine granular patterns [80,84].

Occasionally, high-grade BilINs present a thickened “mucosa” that appears as well-formed propria (fibrovascular connective tissue with inflammatory cells), which appears intramucosal carcinoma of the stomach (Figure 1C) [1]. This may represent intraepithelial involvement of pre-existing metaplastic glands with a propria layer with connective tissue in the bile duct and gallbladder mucosa by neoplastic cells. Otherwise, this may be a unique high-grade BilIN developed by the neoplastic transformation of epithelial lining that accompanies mucosa propria formation.

Neoplastic epithelia of high-grade BilIN involve the non-neoplastic, surrounding biliary mucosa and non-neoplastic glands, including the peribiliary glands and R-A sinus located at various depths from the luminal surface of the biliary tract, as shown in Table 4 [33,71,83,84]. Albores-Saavedra et al. reported the following useful clues for differentiating R-A sinus showing involvement with high-grade dysplasia (in situ carcinoma) from tubular invasive carcinoma [84,85]: (i) connection of the epithelial invagination to the surface epithelium, (ii) recognition of normal biliary epithelia admixed with neoplastic epithelia, (iii) presence of inspissated bile in long dilated spaces and (iv) lack of invasion into the smooth muscle bundle [79,84,85].

#### 3.1.3. Grading

The differences between low-grade and high-grade BilINs are shown in Table 3 [1,4,24,71]. High-grade BilIN can present with fully developed intraepithelial carcinomatous changes. Low-grade BilINs usually involve focal areas, whereas high-grade BilINs form over considerable areas of the biliary tract mucosa [1,71]. Low-grade BilINs are not regarded as direct precursor and are difficult to objectively define [22]. However, high-grade BIlINs are often accompanied by low-grade [22]. Moreover, high-grade areas are mainly found in the central areas, whereas low-grade areas are typically found in the peripheral parts, thus suggesting the possible transition of low-grade BilIN to high-grade BilIN [21,71]. Only high-grade BilIN areas were also identifiable.

#### 3.1.4. Four Subtypes

As for the subtypes based on cell lineage, a majority of high-grade BilINs are of the PB subtype, followed by the gastric and intestinal subtypes [4,71,82]. The characteristic features of the four subtypes are shown in Table 1 [22,71,82]. High-grade BilIN of the oncocytic type has not yet been reported thus far.

#### 3.1.5. Invasion

The following two findings suggest the direct development of stromal invasion from high-grade BilIN.

(i)High-BilIN associated with focal stromal invasion

Occasionally, high-grade BilINs show extensive continuous intraepithelial superficial spreading involving several parts of the bile duct and gallbladder, and some of these cases are actually associated with foci of invasive carcinoma at some point(s) of the biliary tract at the time of surgical resection [81,82,86,87]. These cases may radiologically present with bile duct strictures or dilatation [86].

Practically, careful observation and preparation of many histologic sections are mandatory for surgically resected bile ducts and gallbladder with high-grade BilINs for the detection of stromal invasion [18,22,80,84], particularly in cases without grossly visible CCA or GBC.

(ii)High-grade BilIN in the surrounding mucosa around invasive CCAs and GBC.

High-grade BilINs are frequently detected in the biliary mucosa around TBCs [22,71,81,82]. Moreover, 32% to over 80% of invasive GBCs show areas of high-grade BilINs in the surrounding gallbladder mucosa around the GBC, and high-grade BilINs are also frequently found in the surrounding bile duct mucosa of CCAs [82], suggesting that components of precursor high-grade BilIN remain after the development of BTCs [18,22,81,82].

#### 3.1.6. Origin in the Biliary Tract, Pathogenesis, and Molecular and Genetic Changes

(a)Origins in the biliary tract

Most high-grade BilINs may arise on the mucosal surface of the biliary tract [6,24,82]. In addition, several reports have indicated that high- and low-grade BilIN lesions in considerable areas are recognizable in the R-A sinus [71,80,82,84], adenomyomatous hyperplasia of the gallbladder, and proliferated peribiliary glands of the biliary tract wall [6,28,38] when few or no high-grade BilINs occur on the luminal surface. Certain high-grade BilINs may secondarily extend continuously to the mucosal surface of the gallbladders and bile ducts via conduits of these non-neoplastic glands or cysts [83].

(b)Pathogenesis
Background lesions

Reports have suggested that chronic biliary inflammation and injuries may induce neoplastic changes in the biliary epithelium resulting in the development of BilINs [1,6,22]. CCA and GBC develop a multistep process involving high-grade BilIN beginning with transformed biliary epithelial cells or stem/progenitor cells [23,36]. BilINs reportedly share histopathologies and phenotypes with pancreatic intraepithelial neoplasms (PanINs) [12,24], suggesting a common developmental process and pathological significance with frequent common molecular expression [11]. However, genetic comparisons between BilIN and PanIN have not yet been analyzed in detail.

Field cancerization, which is the replacement of the normal cell population by a cancer-primed cell population, is now recognized as a mechanism underlying the development of many types of cancer, including common carcinomas of the lung, colon, prostate and urinary tract [88,89]. High-grade BilINs usually involve a considerable area of mucosa of the biliary tract, when detected, suggesting the participation of field cancerization [89]. Biological mechanisms that drive the evolutionary process that results in the formation of field cancerization and the development of high-grade BilIN, remain speculative [89].


2.Molecular and genetic changes


High-grade BilINs, CCA, and GBC are inevitable consequences of aging and mutation accumulation [90]. The mean age at the time of detection was 45 years for low-grade BilIN, and 60 years for high-grade BilIN, and 71 years for GBC [90]. High-grade BilINs show sequential molecular and genetic alterations related to cell kinetics [91]. p21, p53, and cyclin D1 expression along with down-regulated DCP4 and p16 expression are observed in the histological progression of BilIN [63,92,93]. Evidence suggests that the expression of autophagy-related protein is also up-regulated at an early stage [94,95] and SMAD4 and glucose transporters are involved in the carcinogenesis of BilIN [93]. Overexpression of the polycomb group protein EZH2 may induce hypermethylation of the P16^INK4A^ promoter followed by a decrease in the expression of p16 in the multi-step cholangiocarcinogenesis of BilIN in hepatolithiasis [96]. A few studies that have performed single gene analyses suggest that *KRAS* mutations occur in approximately 33% of BilIN cases with concomitant iCCA, and these mutations have been identified as an early molecular event during the progression of BilIN, with *TP53* mutation representing a late molecular event [97]. Moreover, the progression of BilIN is accompanied by the upregulation of CD15 [98]. Recently, Loeffler et al. [99] showed that the clustering of 49 deregulated miRNAs corresponded to the three stages of dCCA, thus supporting the concept of BilIN as a tumor precursor, and they identified miR-451A and miR-144-3P as putative tumor suppressors that attenuate cell migration. Mutations in *CTNNB1* have been frequently observed in co-existing BilIN and GBC [100]. Loss of heterozygosity (LOH) has been detected in both BilIN and gallbladder cancerous lesions [101,102,103]. LOH in the 3P14.2 locus has been implicated in the pathogenesis of gallbladder carcinogenesis by inactivating the fragile histidine triad (FHIT) tumor-suppressor gene [104].

Recently, Nagao et al. [105] developed a mouse model and reported that concurrent activation of the *KRAS* and *WNT* pathways induced biliary neoplasms resembling ICPN and BilIN, and these precursors infrequently progressed to invasive carcinoma. In this model, the genes associated with the *C-MYC* and *TGFβ* pathways were shown to be increased.

Lin et al. [90] proposed two evolutionary paths in the development of GBC: the BilIN-independent path, and the BilIN-dependent path. They suggested an evolutionary relationship between carcinoma and low- and high-grade BilINs based on SNV data. The BilIN-independent model is more aligned with the de novo development path, in which initiates before the divergence of low-grade BilIN and high-grade BilIN, and evolves more independently. In contrast, the BilIN-dependent path is similar to the adenoma/dysplasia-carcinoma sequence, in which carcinoma evolves from low-grade BilIN/high-grade BilIN in a step-wise progressive manner. Among individual chromosomes enriched for LOH events, the LOH events on chr.5q and chr.7p have been associated with early carcinogenic changes in the gallbladder [106].

#### 3.1.7. Staging and Prognosis

(a)Staging

High-grade BilINs are categorized as Tis (carcinoma in situ) according to the TNM classification [1,42,43].

The T category parameters for the biliary tract were extrapolated directly from the gastrointestinal (GI) tract [42]. However, in the biliary tract, the muscularis mucosa, which delineates Tis from T1, is absent and the muscular layer is lacunar in the gallbladder. Therefore, Road et al. proposed that designating in situ or minimally invasive carcinomas limited to the muscularis or above as early gallbladder carcinoma may help to alleviate the major geographical differences or difficulties in the criteria for invasion to pTis or pT1 [22,107].

(b)Prognosis

Prognostic studies on high-grade BIlINs without invasive carcinoma of the biliary tract based on many cases are lacking. In many reports that have performed long-term post-operative survival and recurrence analyses, cases with BilINs including high-grade BilINs, at the surgical margin do not present significant differences compared with cases without BilINs at the margin, and such cases do not require additional surgery [72,73,74,108]. These data suggest that high-grade BIlINs in the bile ducts-remain in the body for a long time, suggesting that further events are necessary for high-grade BilINs to initiate the invasion and progression.

Early gallbladder cancer data including many cases of high-grade BilIN suggest that most cases of high-grade BIlINs in the gallbladder without GBC can be cured by cholecystectomy [107]. However, a small portion of patients experience recurrence and metastasis several years after diagnosis, thus indicating a field cancerization effect in the gallbladder [107].

#### 3.1.8. Related Diseases and Differential Diagnosis

(1)IPNB and ICPN

The practical differentiation of high-grade BilINs with micropapillary patterns from ICPN and IPNB has not yet been discussed in detail, although the former has been shown to be microscopically identifiable, while the latter is grossly visible, mass-forming or polypoid, papillary lesion [1,7,40]. The height of high-grade BilIN is typically less than 3 mm [109]. A large size exceeding 1 cm was first proposed by Adsay et al. [18] as an inclusion criterion for ICPN, and a height greater than 0.5 cm from the surrounding mucosa level as an inclusion criterion for typical IPNB [109,110]. Intraepithelial papillary neoplasms higher than high-grade BilINs but lower than 0.5cm could represent incipient IPNB or ICPN (see below).

(2)“Intraductal spread of invasive carcinoma (cancerization)”

In the pancreas, invasive ductal adenocarcinoma is reported to invade back into and extend along the ductal system, thereby morphologically mimicking intraductal neoplasia such as PanIN or even IPMN (intraductal spread or cancerization of invasive carcinoma) by invasive carcinoma [76,77]. It is sometimes impossible to definitely distinguish these carcinomas from high-grade PanIN, because both consist of cytologically malignant epithelia within a duct.

Such cancerization may also occur in the biliary tract in invasive BTC. As for the differentiation of high-grade BilIN from cases of “cancerization of biliary tract” by invasive carcinoma, we tentatively propose the following method of differentiation [71,82]; when these intraepithelial atypical lesions are associated with an underlying infiltrative carcinoma with desmoplastic reaction, they could reflect cancerization of infiltrating carcinoma, whereas when these atypical lesions are not associated with such underlying infiltrative carcinoma in the walls, they may reflect the remaining or preceding in situ neoplasm, particularly, high-grade BilIN, followed by the development of invasive carcinoma [82]. While this proposal must be verified in further studies and validated, an analysis of bile duct cancerization phenomena may support the evaluation of biliary tract carcinogenesis via high-grade BilINs.

(3)Reactive atypia

Regenerative changes with atypical cytoarchitectural features are relatively common in biliary tract mucosa with cholelithiasis or chronic inflammatory changes [1,22,61,66]. The presence of intraepithelial neutrophils, intercellular clefts, and fine and pale nuclear chromatin which favor reactive changes, helps to solve this problem. In addition, in comparison with reactive changes, high-grade BilIN show cytological and structural alterations (see above) [1]. Reactive changes are usually patchy or focal and shows a gradual transition to the adjoining epithelia. In addition, neoplastic cells of high-grade BilINs are frequently, strongly, and diffusely positive for s100-P and p53 [83]. However, Albores-Saavedra et al. reported that pathologists must rely on conventional histological sections to differentiate high-grade BilINs from reactive atypical epithelia [80,84,85].

### 3.2. IPNB

IPNBs are proposed as an overarching category for intraductal, grossly visible (polypoid, papillary or villous), preinvasive neoplasms arising in the bile duct mucosa and growing in the lumen with common overlapping histology [1,7,109,110]. Although several nomenclatures have been applied to these tumors such as biliary adenoma, papilloma, papillomatosis, non-invasive papillary carcinoma and mucin-secreting biliary tumor [1,5,111,112,113], these obsolete terms are not recommended in the current WHO tumor classification [1]. However, when some cases with unique genetic, molecular and clinicopathologic features are established as a single disease, the disease may be separated from “IPNB” and established as a unique preinvasive biliary neoplasm. IOPN of the bile duct may be a candidate [114]. IPNBs are usually found in the extrahepatic and intrahepatic bile ducts or their main branches or cysts that communicate with these bile duct [1,7]. IPNBs frequently involve and spread continuously to more than one anatomical segment of the bile ducts [7,38].

#### 3.2.1. Clinical Features and Background and Epidemiology

IPNB is typically diagnosed in middle-aged or elderly adults and has a slightly higher occurrence in males [7,8,109,110,115,116,117,118,119]. IPNB occurs worldwide, but its incidence varies geographically [7,8,114]. IPNBs account for 10–38% of all bile duct tumors in eastern Asia but only 7–12% of all bile duct tumors in North American and European countries., and a higher incidence is noted in South-East and Far-East Asian countries [7,8,119,120,121]. Risk factors include hepatolithiasis, liver fluke infections, primary sclerosing cholangitis (PSC), congenital biliary tract diseases and exposure to chemicals such as chlorinated organic solvents [7,8,122,123,124]. These usually occur as single tumor and/or occasionally as multiple tumors, and can present clinically as large duct obstructions with recurrent abdominal pain, cholangitis and cholestatic hepatic dysfunction [7,8,115,123].

#### 3.2.2. Gross and Microscopic Features and Intraepithelial Spread and Glandular Involvement

Almost all IPNBs show grossly visible intraluminal neoplastic growth as well as micro/short papillary or flat intraepithelial neoplastic growth to a variable extent in the surrounding bile duct mucosa. IPNB is characterized by these two components [1,7].

The affected bile ducts are grossly dilated, filled with a grossly visible exophytic, papillary, villous or polypoid neoplastic growth (range, 1–6 cm) and frequently present mucus hypersecretion [7] (Figure 2A). Their height is, typically at least 5 mm from the adjacent biliary mucosa [7,38,109,110]. In many cases of IPNB, these exophytic lesions are usually conglomerates of higher polypoid and lower or microscopic papillary or flat preinvasive lesions that present coarse, fine granular, or rough mucosa; however, some IPNBs are isolated or solitary without neoplastic involvement of the surrounding mucosa [7,38].

The gross features of IPNBs depend on their anatomical location, excessive mucin secretion, or macro-invasion of the liver [1,7,38]. IPNBs develop similarly in intrahepatic and extrahepatic bile ducts [7,38]. IPNBs located in intrahepatic bile ducts tend to be larger in both mass and extension than in extrahepatic bile ducts [7,38]. Some IPNBs, particularly those arising in the extrahepatic bile ducts, are associated with a cylindrical or fusiform morphology with moderate dilatation of the affected bile ducts, and appear as a cast-like structures, while IPNBs in the intrahepatic bile duct tend to present marked macroscopic dilatation or multilocular cystic changes [7,38,124]. These cystic changes usually show luminal communication with the adjacent bile duct, as shown on imaging. However, anatomical communication between these cystic lesions and the bile duct is sometimes difficult to confirm pathologically. These cysts usually contain papillary neoplastic components and mucinous fluid in variable proportions, and their internal surfaces are smooth or finely granular, implicating intraepithelial involvement. Interestingly, IPNBs arising in the peribiliary glands or smaller bile ducts directly branching from the hilar bile ducts show saccular or aneurysmal dilatations attached to the hilar bile duct [125,126,127].

Mucin hypersecretion is found in approximately half of the IPNB cases [38,128]. Mucin hypersecretion is more common in intrahepatic IPNB than in extrahepatic IPNB.

Grossly visible lesions generally have polypoid, papillary or villous basic histopathology and show intraductal growth composed of single-layered or pseudostratified neoplastic biliary epithelia covering fine fibrovascular stalks that lack an ovarian-mesenchymal type stroma in the dilated bile duct lumen [38]. Fibrovascular stalks can be edematous or inflamed. While some appear predominantly villous or papillary, others have predominantly tubular or papillo-tubular patterns with limited intervening connective tissue (back-to-back epithelial units) depending on the four subtypes of neoplastic epithelia [7,38].

In addition, many cases of IPNBs show intraepithelial involvement of the surrounding mucosa (lateral spreading) and the peribiliary glands in the bile duct walls (Table 3). A higher grade is more common near the main intraductal neoplasms, and the peripheral parts are more differentiated or low-grade. Both grossly visible neoplasms, intraepithelial lateral spreading and glandular involvement constitute IPNB [1,7,38]. However, some IPNBs are not associated with the surrounding intraepithelial neoplastic lesions and can progress differently.

#### 3.2.3. Pathological Grading: Modified Two-Tiered Grading (Type 1 and Type 2 Subclassification)

IPNBs are traditionally, pathologically graded as low-grade dysplasia (LGD) and high-grade dysplasia (HGD) according to cytoarchitectural alterations [38,75]. A majority of IPNBs belong to HGD and have a high potential to progress into invasive tumors, although the criteria for LGD and HGD are subjective among pathologists and this grading does not reflect post-operative survival [38,115,116]. To supplement this traditional grading system, a Japan–Korea study group proposed type 1 and type 2 subclassifications (modified two-tiered grading system) [109,110] (Table 5).

This subclassification is based on two characteristic features of IPNB: (i) similarities with other precursors of the pancreatobiliary system, for example, prototypes of IPMN and IOPN of the pancreas and those of intestinal adenoma of the ampulla, and (ii) unique cytoarchitectural alterations, for example, type 1 IPNB (gastric, PB and oncocytic subtypes) shares many features of prototypes of gastric and PB subtypes of IPMN and those of IOPN with regular histology, and type 1 IPNB (intestinal subtype) shares many features of low- and high-grade intestinal adenoma of the ampulla with regular histology, respectively (Figure 2B), while type 2 shows variable differences from the gastric and PB subtypes of IPMN and IOPN of the pancreas [7] and intestinal adenoma of the ampulla [1] (Figure 2C). In addition, type 2 IPNB often shows complicated structures such as cribriform and solid patterns and bizarre cells or nuclei, thus suggesting overt malignancy, and coagulative necrosis, whereas type 1 rarely presents such complicated features [1,7]. Adsay et al. also reported that a small subset of HGD in ICPN showed irregular structures, such as a cribriform arrangement and solid areas, and even exhibited comedo-like necrosis [18], thus corresponding to the above-mentioned type 2. In almost all cases of type 2, a variable amount of LGD or type 1 lesions, at least in small areas, are found or admixed.

Approximately 40% of IPNBs are reportedly type 1, while the remaining 60% of IPNBs are type 2 [7,10,38,110,129]. Mucin hypersecretion is more common in type 1 (61%) than in type 2 (37%) [38]. MUC1 expression is more frequent in type 2 (100%) than in type 1 (59%) [7]. Recent studies have shown that this modified two-tiered grading represents a promising approach for assessing post-operative prognosis in IPNB [109], whereas traditional grading does not reflect post-operative survival [38]. That is, type 1 was associated with favorable post-operative outcomes in comparison with type 2 [109], thus indicating that this modified two-tiered grading is clinically applicable in the prognostic evaluation of IPNBs.

#### 3.2.4. Subtypes Based on Cell Lineage

Characteristically, IPNB presents predominantly as one of the four subtypes, while an admixture of two or more subtypes is not infrequent. These four subtypes show unique histological features according to their grading [7,38].

(i)Intestinal subtype

The intestinal subtype of IPNB is composed of papillary/villous, tubule-villous, tubular, and tubular patterns [7,38,109,130,131]. The papillary pattern may correspond to a “transformed papillary configuration of the tubular pattern which is frequently observed in tubular intestinal adenomas. The serrated type was also focally detected in the intestinal subtype of IPNB. In type 1, these structures showed regular lesions, whereas in type 2, irregular, heterogeneous lesions were observed. One-third of IPNBs of the intestinal subtype were mainly composed of the villous/papillary pattern (more than two-thirds), whereas one-fourth of IPNBs mainly presented the tubular pattern (more than two-thirds) [7,38]. More than half of IPNB cases were composed of mixed papillary/villous and tubular components (each accounting for more than one-third) [7,38].

This intestinal subtype pattern of IPNB is different from that of IPMN, which is exclusively composed of villous patterns and the tubular pattern has not been reported, and it is very similar to that of intestinal adenoma of the colorectum and ampulla, which show tubular, villous or tubule-villous patterns. In this context, the intestinal subtype of IPNB could be a counterpart of intestinal adenoma of the ampulla, but not of the intestinal subtype of IPMN.

(ii)Gastric subtype

The basic structures showed neoplastic areas that resembled the gastric foveola and pyloric glands in variable proportions [7,38,40]. The foveolar area was composed of columnar to low columnar cells with basally oriented, single-layered or pseudostratified nuclei and abundant supranuclear pale vesicular mucinous cytoplasm. The pyloric gland components were arranged as glandular structures with cuboidal or low columnar epithelia, basally located nuclei and abundant clear supranuclear cytoplasm. In type 2, the two component patterns were irregular or immature, although the transition between foveolar and pyloric gland region may be controversial. One-fourth of IPNBs of the gastric subtype predominantly showed the pyloric-gland-like pattern, half showed predominantly the foveola-like pattern, and one-fourth of IPNB showed a mixed foveola-pyloric gland-like pattern.

Relation to pyloric gland adenoma of the bile duct: While several studies have reported PGA in the bile duct and pancreatic duct, recent pathological, immunohistochemical and genetic studies suggest that such PGAs could be included in the spectrum of the gastric subtypes of IPNB and IPMN, particularly those showing predominant pyloric gland components [7,38,40,131,132,133].

(iii)Pancreatobiliary subtype

IPNB of the PB subtype typically shows many fine papillary structures with numerous fine ramifying branches of thin fibrovascular stalks [1,7,38]. In type 1, such structures are homogeneous and regular throughout the tumor, whereas in type 2, these papillary architectures are irregular and, usually lined by single-layered or mildly stratified medium to large cells with hyperchromatic nuclei and prominent nucleoli.

(iv)Oncocytic subtype

Almost all IPNBs of the oncocytic subtype show arborized papillary and/or cribriform formations [1,4,7,39,134]. In type 1, “papillary lesions” that showed single to multi-layered lining of epithelia on fibrovascular stalks or “compact growth” of neoplastic epithelia with regular secondary lumina were observed. The neoplastic lining epithelia are medium-sized cells with a relatively substantial, oncocytic and finely granular cytoplasm, and with small nuclei and small nucleoli. The foci of neoplastic epithelia of the gastric subtype are commonly observed. In type 2, irregular “papillary lesions” and “compact” larger cells and nuclei with prominent nucleoli were observed.

Intraductal oncocytic papillary neoplasm (IOPN): Fusions involving *PRKACA* and *PRKACB* genes have been detected specifically in oncocytic IPMN and oncocytic IPNB [114]. Oncocytic IPMN and oncocytic IPNB also show similar histology and phenotypes [134]. Accordingly, they are termed intraductal oncocytic papillary neoplasms in the pancreas and described separately from other IPMNs [1,114]. Thus, oncocytic IPNBs can be described as IOPNs differentiated from IPNBs.

Frequency of the four subtypes along the biliary tree: The frequency of the four subtypes varies based on the location along the bile duct and across geographical regions [7,38]. An analysis of 186 cases of IPNB in the intrahepatic and extrahepatic bile ducts showed that the intestinal type was the most frequent IPNB (46%) followed by the gastric (21%:) and PB subtypes (17%) [7,38], with the oncocytic subtype being the least common (16%). In the intrahepatic bile duct, the intestinal type accounted for 37% followed by the gastric subtype (27%), oncocytic type (23%) and PB type (13%), while in the extrahepatic bile duct, the intestinal type accounted for 58% followed by the PB type (21%), gastric type (14%) and oncocytic type (6%). However, Wan et al. examined a Chinese series and reported that 90% of IPNBs were classified into biliary and intestinal subtypes [135], suggesting different histological criteria for each subtype or geographical differences in occurrence.

Rarely, neuroendocrine carcinoma is admixed with the intraluminal components of IPNB (mixed neuroendocrine and IPNB) [136].

#### 3.2.5. Invasion and Multiple Occurrence

(1) Invasion

While the pathologies of the intraluminal components of IPNB are well described [1,7], the pathologic spectrum of invasion remains to be clarified. At the bottom of inraluminally growing papillary neoplasms, which usually present with type 2 features, foci of invasive carcinoma are found dripping into the mucosa. Evidence of adenocarcinoma invasion into the walls and periductal tissue of the bile duct and neighboring organ(s), particularly the hepatic parenchyma, has also been observed [7,38]. Invasion may also start at the bottom of the surrounding intraepithelial neoplastic components away from the main papillary neoplasm [7]. In addition, some cases show invasive growth within the intraluminal preinvasive neoplastic components of IPNB, thus occupying considerable areas and appearing as nodular growth with a desmoplastic reaction. Infrequently, non-invasive papillary components of IPNB are found around and at the periphery of invasive carcinoma. Such cases could be termed “invasive carcinoma with IPNB components” such as in other organs, for example., the colorectum (adenocarcinoma with adenoma components) [1].

Invasive carcinomas arising from IPNB are usually tubular adenocarcinomas with mostly PB features [7,38]. Some cases of IPNB of the intestinal subtype are associated with invasive carcinomas with intestinal features [7]. Interestingly, in the oncocytic subtype, invading carcinoma presents as oncocytic adenocarcinomas and the same genetic and molecular alterations as the intraductal oncocytic components [134]. Occasionally, the invasive parts show a considerable amount of mucinous carcinoma (colloid carcinoma) [7,38].

Approximately half of type 2 IPNBs are associated with invasive carcinoma at the time of surgical resection, whereas a limited amount of type 1 IPNB show invasion [7,38]. Extrahepatic IPNB more frequently shows invasive carcinoma than intrahepatic IPNB [137].

(2) Multiple occurrence of IPNB (multiple IPNB)

Multiple IPNB occurrences are occasionally multiple synchronously and non-synchronously observed along the bile duct [137,138,139,140]. Multiple IPNB tumors usually have similar histopathological and phenotypic characteristics, although their grades may differ. The intraluminal dissemination of tumor cells in the bile duct appears to be more likely because the majority of recurrent tumors develop in more distal parts of the bile duct compared with the primary tumor and show higher grade than those near the liver [137,138,139]. However, a widespread neoplastic field defect [88,89] may also occur in the bile duct resulting in multiple carcinomas.

#### 3.2.6. Pathogenesis and Molecular Genetic Alterations

a. Origins in the biliary tract

The majority of IPNBs may arise from the biliary lining cells of the bile ducts, while some may develop via the peribiliary glands, particularly biliary stem/progenitor cells [7,38,124,126,127,140]. Recently, Pedica et al. reported that 4.6% of peribiliary cysts in alcoholic cirrhosis had low-grade IPNB confined to cystically dilated peribiliary glands, suggesting that they could be an incipient IPNB arising in the peribiliary glands [141]. Such cystic and micropapillary lesions affecting the peribiliary glands were also detected in 9 (1%) of 938 consecutive autopsy cases, and could be a source of IPNB [142]. Recently, Tomita et al. [143] reported a mouse model in which IPNB may arise from peribiliary glands (see below).

b. Pathogenesis

(1) Background lesions

While the background lesions of the majority of IPNBs remain unknown, some cases are associated with chronic biliary injuries such as hepatolithiasis and liver fluke infection [7]. In IPNB cases with chronic exposure to chlorinated organic solvents, the solvents may act as carcinogens [122,123,144].

The Histology and immunohistochemical features of the main papillary lesion and surrounding intraepithelial neoplasm were consistent, suggesting that the former might have preceded the latter. Intraductal papillary neoplastic lesions <5 mm in height that fulfill the histological features of IPNBs (incipient IPNBs) may increase in size to that of IPNBs (see below).

(2) Molecular and genetic changes

IPNBs show a multistep progression toward CCA based on molecular and genetic alterations [7,38,129,145]. To date, molecular and genetic alterations that are common in all IPNB cases have not been established. These alterations could differ according to the four subtypes and stages of malignant progression [7,38,129,146].

i. Low-grade, high-grade and invasive IPNBs

IPNBs show stepwise acquisition of molecular alterations that affect common oncogenic pathways, such as cell-cycle related molecules [93,94]. The upregulation of autophagy-related proteins in IPNB in hepatolithiasis suggests a role for dysregulated autophagy at an early stage of IPNB [94]. Overexpression of EZH2, IMP3, and p53 may also be associated with malignant behavior in IPNB and may occur in parallel with up-regulated MUC1 and down regulated MUC6 [147,148].

Recently, Yang et al. [146] and Aoki et al. [129] used targeted next-generation sequencing to identify the frequent mutations in cancer-related genes. For example, mutations in *KRAS* (49%), *GNAS* (32%), *RNF43* (24%), *APC* (24%), *TP53* (24%), *CTNNB1* (11%), *APC* (14.3%), *SMAD4* (14.3%) and *GNAS* (11.4%) were detected in IPNBs [129]. The rate of *KRAS* mutation is significantly higher in high-grade and invasive IPNBs, and this mutation is significantly associated with tumor size and Ki-67 expression [129]. *SMAD4* mutations and loss of SMAD4 protein have only been identified in the late phases of tumor development [93].

ii. Type 1 and type 2 subclassification

Aberrant expression of p53 and loss of SMAD4 were more frequent in type 2 than in type 1 IPNB [129,146], thus reflecting the more malignant and invasive characteristics of type 2 IPNB. Aoki et al. reported that among mutations of genes, mutations in *KRAS* were more frequent while mutations in *GNAS* and *RNF43* were only found in type 1 IPNBs [114]. In this context, type 1 IPNBs share many features with IPMNs [129]. On the other hand, type 2 IPNBs show frequent *TP53*, *SMAD4* and *KMT2C* mutations but rarely show *GNAS* mutations [129,146]. Yang et al. also found that type 2 IPNBs frequently show *P53*, *SMAD4* and *KMT2C* mutations [146]. These genetic studies suggest that IPNBs consist of at least two distinct pathogenetic types based on gene mutations. A higher frequency of overexpression of DNMT1, DNA methylation-catalyzing enzymes, in type 2 (100% for type 2) compared to type 1 (28.6%), suggest that DNA methylation of these suppressor genes may be more involved in the pathological and biological characteristics of type 2 IPNB [130].

iii. IPNB with intestinal and non-intestinal pancreatic IPMN and IOPN

Several reports have identified similar genetic alterations in IPNB and IPMN [129,149,150,151], with some being observed in the same cases [151]. IPNBs of the intestinal subtype share genetic alterations with IPMNs [131]. Among recurrent mutations in IPNBs, Thsai et al. reported that *GNAS* mutations were detected in fewer than half of all cases of IPNB, and that all cases with *GNAS* mutations could be differentiated by villous architecture and mucus hypersecretion [150]. Mutations in *RNF43*, a tumor suppressor gene, are also frequent in intestinal IPNBs [150,152]. When divided based on intrahepatic and extrahepatic location, intestinal IPNBs arising in the intrahepatic bile ducts presented frequent *GNAS*, *KRAS*, and *RNF43* mutations [131], suggesting that intestinal IPNBs, particularly those arising in the intrahepatic bile duct, are the biliary counterpart of IPMN [129,131,150]. However, genetic mutations in *TP53* and *PIK3CA*, which are infrequent or absent in intestinal IPMNs, were frequently detected in extrahepatic intestinal IPNBs [130].

Fujikura et al. reported that mutations in *APC* or *CTNNB1*, both of which belong to the *WNT/β-catenin* pathway, were observed in 43% of IPNB cases, thus resembling non-intestinal subtypes of IPMNs [153]. *APC* and *CTNNB1* mutations and subsequent activation of the *WNT/β-catenin* signaling pathway are sufficient to fully activate the *WNT/β-catenin* pathway and can be used to differentiate IPNBs with the non-intestinal phenotypes. Another report showed that the PB subtype of IPNB arising in the extrahepatic bile ducts also harbored the *CTNNB1* mutation [129,146]. All IPNBs with *CTNNB1* mutations were of the PB subtype, located in the extrahepatic bile duct, and lacked mutations in *KRAS*, *APC*, *RNF43*, and *GNAS* [150,151]. Activation of the WNT/β-catenin signaling pathway associated with *CTNBB1* mutations may be involved in the development and progression of non-intestinal-type IPNBs, particularly the PB subtype [149].

The oncocytic subtypes in the pancreas and bile duct are known to lack *KRAS* mutation [121,135,154]. Recently, the *PRKACA* or *PRKACB*-related fusion gene was detected in all 23 oncocytic tumors investigated in IPMNs and IPNBs and this fusion gene was regarded as a specific molecular event in oncocytic IPMN and IPNB [114]. The mucin profile of oncocytic IPNB was similar to that of its pancreatic counterpart [134]. Another recent study revealed that the onocytic subtypes of IPNB and IPMN showed different expression patterns in several signaling pathways, increased expression of *follistatin (FST)* and lower apoptotic activity relative to those of the other subtypes of IPNB and IPMN [154], suggesting a common molecular signaling pathway in oncocytic IPNB and IPMN [113].

iv. IPNB arising from the peribiliary glands

Tomita et al. [143] recently developed an IPNB induced by autocrine/paracrine growth factor, fibroblast growth factor 10 (FGF10). FGF10 is known to regulate branching and tubule formation in bile ducts [29]. FGF10 induces IPNBs that originate from biliary stem/progenitor cells located in the peribiliary glands, mimic the gastric and PB subtypes, and mimic multifocal and divergent human IPNB phenotypes [143]. The autocrine/paracrine growth factor FGF10 is also involved in mucin hypersecretion in this IPNB model [143]. Interestingly, with *KRASG^12D^*, *P53* or *p16* mutations or a combination of these mutations, FGF10-induced IPNB shows step-wise carcinogenesis, thus causing associated invasive carcinoma. The development and progression of these papillary neoplasms were suppressed by the inhibition of the *FGF10-FGRFR2-RAS-ERK* signaling pathway [143].

#### 3.2.7. Staging and Prognosis

(1) Staging

IPNBs of HGD without invasion were staged as pTis. The staging of IPNBs with associated invasive carcinoma was similar to that of iCCA, pCCA and dCCA [7,42].

(2) Prognosis

The prognosis of IPNB as well as invasive carcinoma derived from IPNB has been consistently better than that of conventional CCA [155,156,157,158,159,160]. Invasive IPNB showed poor survival compared with non-invasive IPNB [1,38]. The prognostic significance of flat or micropapillary intraepithelial neoplasms of the IPNB at the surgical margin remains controversial [7]. In CCA derived from IPNB, the rates of tumor recurrence and overall survival were 47% and 68.8%, respectively, after five years. [1,120,160,161]. Moreover, the difference in post-operative survival with respect to LGD and HGD in IPNB is controversial [1,7,38], with the IPNB of type 1 showing a favorable prognosis in comparison with that of type 2 IPNB [108,128]. Both the depth and percentage of invasive carcinoma components are correlated with poor survival [1,157,162]. Extrahepatic IPNB showed a more unfavorable post-operative prognosis than intrahepatic IPNB [46,138]. Multiplicity is a common feature of IPNB and has a negative effect on IPNB progression [128]. Invasive CCA derived from IPNB with the PB type show higher histological grades, more lymph node metastasis, and more frequent postoperative recurrences [163,164]. MUC1 expressing CCA derived from IPNB show a shorter recurrence-free survival time than MUC1 not-expressing CCA derived from IPNB [1,8,121]. In contrast, colloid carcinomas derived from IPNB have a better prognosis than tubular adenocarcinoma [112,113,155].

#### 3.2.8. Related Diseases and Differential Diagnosis

(1) Related diseases

a. Incipient IPNB

Preinvasive intraepithelial neoplasms <5 mm in height but >3 mm in the bile duct are occasionally encountered in the extrahepatic bile duct without invasive carcinoma and in other parts without IPNB [7,38]. They present with one of four subtypes, usually LGD or HGD, with a low-grade component. While they are lower in height, they are similar to IPNB; therefore, they could be termed incipient IPNB as in the pancreas (incipient IPMN) [75].

b. Microscopic intraepithelial neoplasia with mucin hypersecretion

Several reports have shown extensive bile duct dilatation filled with mucin and lined by superficially spreading, microscopically identifiable, non-invasive biliary neoplasms [119,163,164,165]. However, whether such cases could be included as variants of IPNB with unusual presentations remains to be clarified.

c. Predominantly tubular growth in bile duct

While the pyloric gland predominant gastric subtype and tubular component predominant intestinal subtype are included in IPNB, several cases of predominantly tubule forming polypoid neoplasms with poor or no differentiation into four cell lineages have been reported [166,167,168]. Intraductal tubulopapillary neoplasm (ITPN) is a recently proposed intraductal, grossly visible, polypoid premalignant neoplasms with a predominantly tubular growth pattern [167]. Large intrahepatic and extrahepatic bile ducts are commonly affected, and many cases are associated with invasive adenocarcinoma at the time of diagnosis (approximately 80%) [166,167,168]. The intrahepatic component tends to be more nodular than the cystic components, and a snake-like intraductal growth pattern is often characteristic. The tumor showed high cellularity with back-to-back tubular glands and solid sheets with minimal papillary architecture and frequent necrosis. They expressed MUC6 (30%) but not MUC5AC and lacked intestinal differentiation. Genetic alterations involving *CDKN2A/p16* and *TP53* have been reported in ITPN [169]. Recently, Gross et al. [168] performed whole exome sequencing of ITPN and revealed high genetic diversity with recurrent copy number variants (CNVs) (loss of chromosome 1P36 and others), and only a few recurrent somatic mutations in *TG, SLIT2, FGFR2, and HMCN1*.

(2) Differential diagnosis

a. Adenomyomatous hyperplasia of the bile duct

Adenomyomatous hyperplasia (AH) is a rare tumor-like inflammatory hyperplastic lesion with bile duct strictures [169,170,171,172]. Multiple, prominent biliary epithelial proliferations with tubular-papillary architecture and minimal nuclear atypia are associated with chronic inflammation, fibroblastic proliferation and smooth muscle proliferation. Polypoid AHs of the extrahepatic bile ducts resemble to those of IPNB [119,172]. However, the characteristic inflammatory and fibrotic reaction and regenerative features of proliferated epithelia may differentiate AH from IPNB [172].

b. Polypoid invasive carcinoma (PICA)

PICA is a CCA showing intraluminal polypoid growth, and is occasionally present in the bile duct [173]. All intraductal polypoid components and invasive parts of the wall were composed of invasive adenocarcinoma, and few or no adenomatous components were found in the intraluminal polypoid components. In contrast, in IPNB associated with invasive carcinoma, preinvasive or low-grade components are identifiable in the intraluminal polypoid or papillary areas, which differ from the features of PICA. However, IPNBs with invasive nodular growth within the intraluminal components and more advanced IPNBs with a peripheral rim of non-invasive papillary components may be confused with PICA.

c. Metastasis carcinoma and cancerization of small iCCA

Metastatic carcinomas, particularly from colorectal carcinoma, present intraductal papillary growth in the bile duct. Small duct iCCA can present with cancerization that appear as polypoid tubular adenocarcinoma in the dilated intrahepatic large bile duct [10,174]. In these cases, the surrounding bile duct epithelia adjacent to the tumor appeared normal, which could be a clue for differential diagnosis from IPNB.

### 3.3. ICPN

ICPNs are a recently proposed disease entity of the gallbladder, and they cover overarching categories or histologies for grossly visible, polypoid, mass-forming, preinvasive neoplasms arising from the mucosa and growing in the gallbladder lumen [1,9,18,175,176]. Several nomenclatures have been previously applied to these tumors according to the dominant feature(s) such as tubulopapillary (tubulovillous) adenoma, non-invasive papillary neoplasm/carcinoma, papilloma and papillary carcinoma [1,130,177,178,179,180,181]. A substantial overlap of morphology and cellular differentiation is often observed among these cases, although there are no specific clinical implications for their subcategorization, and the use of these obsolete terms is not recommended. However, pyloric gland adenoma (PGA) has been described separately from ICPN in current WHO terminology [1]. In comparison with pancreatic IPMN and IPNB, the clinicopathological characteristics of ICPN are poorly understood [18,176,181]. “Intracholecystic papillary-tubular neoplasm” was originally used for ICPN [1], although the WHO used the same acronym for “intracholecystic papillary neoplasm” in 2019 [1].

#### 3.3.1. Clinicopathological Features and Background and Epidemiology

ICPNs are twice as common in females as in males, and the patients range in age from 20 to 94 years (mean: 61 years) [1,18]. Almost half of all patients with ICPN present with right upper outer quadrant pain. Almost half of these cases are radiologically diagnosed as gallbladder cancer [18]. Approximately 6% of gallbladder carcinomas that arise in association with ICPN [1,18,176,178]. Approximately a half of ICPN cases are associated with invasive carcinoma (ICPN associated with an invasive carcinoma) at the time of surgical resection [18,176]. ICPN is found in 0.4% to 1.5% of all cholecystectomies, and its incidence ranges from a rare disease to 23.5–28% of primary epithelial gallbladder neoplasms [18,176,178], though there have been no systematic geographical reports on its incidence of ICPN. There are no known etiologic factors in ICPN [1], although several biliary diseases such as pancreatobiliary malunion, xanthogranulomatous cholecystitis, lymphocytic cholecystitis, and segmental adenomyomatosis are associated with ICPN [18,176,182,183,184]. Cholecystolithiasis is found in 20% of the ICPN cases [18].

#### 3.3.2. Gross and Microscopic Features and Intraepithelial Spreading

##### Gross and Microscopic Findings

ICPNs are characterized by granular, friable excrescences or prominent exophytic growth on the gallbladder mucosa (Figure 3A). Granular excrescences are usually sessile and broad-based, although they may have thin stalks, and the lesions often detach from the mucosal surface [1,18,176]. Although ICPNs can be located in any part of the gallbladder, they frequently involve several parts, particularly the body and fundus [18,176]. Recent reports have identified ICPN arising in the cystic duct of the gallbladder [185,186]. Adsay et al. reported ICPNs with the largest size of >10 mm [18], and this size criterion has been used as an inclusion criterion for ICPNs in many reports [178,182,183]. The median tumor size is 2.2 cm (range, 1.0–7.7 cm) [18], and the height of approximately half of the ICPN cases ranges from 5 mm to 10 mm, with the remaining over 10mm [176].

ICPNs are grossly classifiable into conglomerated type and solitary types (Table 6): The former accounts for approximately three-fourths of ICPNs and appears to be multifocal or coalescent, while the latter accounts for approximately about one-fourth of ICPNs and is solitary and usually single [176]. The former is usually sessile or has wide stalks, and it is poorly demarcated from the surrounding non-neoplastic mucosa. The lesions around the main grossly visible lesions appear rough, or finely or coarsely granular [176]. Based on meticulous microscopic examination, multiple lesions may be shown to be continuous. In contrast, the solitary type is polypoid and single and it is pedunculated, and usually well demarcated from the surrounding mucosa. The surrounding mucosa was both unaffected and normal.

Histologically, ICPNs more commonly display morphological heterogeneity and diversity than their counterparts, IPMNs and IPNBs [1,18,176]. Mass-forming or grossly visible main lesions of almost all ICPNs show a mixture of papillary/villous areas with fine fibrovascular stalks and tubular areas with minimal intervening stroma (back to back epithelial units) (Figure 3B) [1,18,176]. Among them, 43% were mostly papillary, 26% were mostly tubular, and 31% were tubulopapillary [18]. The papillary and tubular structures of ICPNs are histologically lined by single layered or pseudostratified cuboidal and columnar epithelia. Inflammatory reactions and edematous changes are common in these stalks and stroma.

In the conglomerated type, neoplastic epithelia of ICPN show intraepithelial spread and glandular involvement (Table 4) [176]. That is, the adjacent mucosa around the main tumor(s) often reveals flat and micropapillary intraepithelial neoplasms to a variable extent, which correspond to ICPNs with surrounding dysplastic changes [18,176]. Occasionally, this spread to the gallbladder mucosa is extensive. In addition, intraepithelial neoplastic epithelia involving R-A sinus and other non-neoplastic components in the walls, appearing invagination or pseudo-invasions [18]. Both grossly visible and intraepithelial lateral spreading and glandular involvement are considered constituents of ICPN, and the histogenesis of these main tumors and the involvement of intraepithelial lateral or glandular components may be the same [176].

In contrast, for the solitary type, intraepithelial growth or intraglandular involvement is very limited or absent and corresponds to ICPN without surrounding dysplastic changes [175,185]. Certain ICPN cases arising in the cystic duct, show extensive intraepithelial spread in the extrahepatic bile duct (conglomerated type), whereas some show limited growth in the cystic duct (solitary type) [186].

Mucus hypersecretion has been also reported in ICPN. However, the incidence of mucus hypersecretion varies from 10.5% to 43% in ICPN cases [175,176,187], probably due to the different diagnostic criteria for ICPN.

As a subset of ICPN, intracholecystic tubular non-mucinous neoplasm (ICTN) of the gallbladder was proposed by Pehlivanoglue et al. [182] and may be solitary type of ICPN [176]. ICTN is characterized by small, non-mucinous tubular units with a complicated architecture [182]. These lesions are always pedunculated with thin stalks. Microscopically, they have a cauliflower-like architecture and are composed of compact, back-to-back acinar-like units with minimal or no cytoplasm. Squamoid/meningothelial-like morules were often observed. The scant stroma sometimes exhibits amorphous amyloid-like hyalinization. Cytologically round nuclei with single prominent nucleoli are prototypes, and they are usually positive for MUC6 but negative for MUC5AC. All cases were regarded as HGD, and ICTNs were not associated with invasion. ICTNs are associated with *WNT/β-catenin* pathway alterations [182], and the background mucosa typically does not show any dysplasia [167]. Therefore, a consensus on the differentiation of ICTN from PGA is necessary.

#### 3.3.3. Pathological Grading: Two-Tiered Grading System

Based on the two-tiered grading system, ICPNs are pathologically graded as low- or high-grade lesions according to the highest degree of cytoarchitectural atypia applied to BilIN [1,18,75,176]. Almost all ICPNs belong to HGD [18,75,176], and contain more atypical or high-grade areas than IPNB and IPMN. The majority of high-grade ICPN contains variable amounts of low-grade components (Figure 3C), and transition from LGD to HGD in ICPN is evident in many cases [18,176]. ICPNs composed entirely of LGD are rare [188]. As in IPNB, type 1 and 2 subclassifications [7,38] are also applicable in ICPN, with one third- and two-thirds of ICPNs subclassified into types 1 and 2, respectively [176]. Stromal invasion was frequent in type 2 (54.5%) but rare in type 1 (12.5%). All ICPN cases with mucin hypersecretion were type 1 and showed no invasion [176].

#### 3.3.4. Subtypes Based on Cell Lineage

The classification of ICPN into four subtypes based on their predominant features has been aided by immunohistochemistry [1]. Compared with IPMN and IPNB, the differentiation may be incomplete or immature, and mixed or hybrid cases are frequently present, and unclassifiable areas are also common in ICPNs [18,176]. ICPNs tend to have a high cell lineage diversity [18,176]. However, they are classifiable into the mixed subtype (more than two subtypes are predominant) and mono-subtype (one subtype is predominant in the whole tumor (>75%)) [176]. In the mono-subtype (63% of ICPNs), the characteristic features of the four subtypes are similar to those of IPNB and IPMN, although they were more complicated and diverse in ICPN. The mixed type (37%) was characterized by a considerable mixture of more than two subtypes. The gastric, PB and intestinal subtypes are relatively frequent, whereas the oncocytic subtype is occasional [18,176].

(i) The biliary subtype is frequent, and 50% of all ICPNs [18] and 17% of the mono-subtype belong to this subtype [176]. This type is lined by cuboidal cells showing a clear to eosinophilic cytoplasm, enlarged nuclei, and prominent nucleoli. Patients with the biliary subtype had the highest risk of concurrent adenocarcinoma.

(ii) The gastric subtype is frequent, and 44% of all ICPNs [18] and 46% of the mono-subtype belonged to this subtype [176]. This type is reflected in elongated glands lined by tall columnar cells with abundant pale cytoplasm and peripherally located nuclei (gastric foveola type). Moreover, it is characterized by smaller tubular glands lined by relatively uniform cuboidal cells with a modest amount of cytoplasm, round nuclei, and visible nucleoli.

(iii) The intestinal subtype is frequent and 8% of all ICPNs [18] and 29% of the mono-subtype belonged to this subtype [176]. This subtype closely resembles that of colorectal adenoma and presents tall columnar cells showing pseudostratified cigar-shaped nuclei and a basophilic cytoplasm.

(iv) The oncocytic subtype is occasionally observed and 8.6% of all ICPNs [18] and 8% of the mono-subtype belongs to this subtype [176]. This type is characterized by arborizing papilla lined by multiple layers of cells with abundant acidophilic finely granular cytoplasm and single prominent nucleoli.

In addition, there are occasional cases of mixed neuroendocrine-non-neuroendocrine neoplasm (MiNEN) arising in ICPN [189]. In this case, adenocarcinoma and large cell neuroendocrine carcinoma were associated with ICPN, and these three components showed similar genetic alterations, suggesting a monoclonal origin.

#### 3.3.5. Invasion

Invasive carcinoma was identified in approximately half of all ICPNs at the time of surgical resection [1,18,82,175,189,190]. Among the GBCs, 6.4% were associated with ICPNs, suggesting that these GBCs might have developed from ICPN. These ICPNs may progress to conventional nodular or flat sclerosing GBC with variably sized visible intraluminal papillary components [18,176], although whether ICPN may progress to PICA has not been clarified [180]. At the bottom of the main parts of the ICPN, invasive carcinoma in the mucosa is continuous with the intraluminal ICPN. Invasive carcinoma was also found to be discontinuous in the muscle layer and subserosa. Invasion can also develop at the micropapillary or flat intraepithelial neoplastic lesion apart from and around the main papillary tumor (4% of ICPNs) [18,176]. Occasionally, invasion is also identifiable in the intraluminal polypoid components (9% of ICPN) [18]. The invasive component is often a tubular adenocarcinoma of PB features with variable desmoplastic reaction, although other types, such as mucinous carcinoma and NEC, have also been described [18,190].

Extensive HGD, predominance of non-pyloric gland cell lineage, PB morphologies or subtype, and mixed subtype were associated with a high risk of invasion in ICPN [1,18,176,189]. Almost all cases of type 1 ICPN did not show invasion, while more than half of type 2 cases with irregular histology and complicated lesions presented invasion [175]. Stromal invasion is relatively more frequent in the conglomerated type than the isolated type [83,175]. The mono-subtype was common in ICPNs with no invasion, whereas the mixed-subtype was relatively associated with invasion [176].

In the gallbladder, the muscularis mucosa is absent and the muscular layer is lacunar. In addition, frequent non-neoplastic glands and R-A sinus continuous with the lumen of the gallbladder were observed in the wall. Therefore, such peculiar structures of the gallbladder may allow neoplastic epithelia for intraepithelial involvement of non-neoplastic glands or R-A sinuses in non-invasive ICPN, and these features are similar to but different from those of invasive carcinoma [18,176]. ICPNs can be differentiated from invasive carcinoma according to the intraepithelial involvement of these non-neoplastic tubes or the R-A sinus [1,18,176], which is similar to the observation for high-grade BilIN (see above).

#### 3.3.6. Pathogenesis and Molecular and Genetic Changes

a. Origins in the biliary tract

Most of ICPN arise from the surface mucosa of the gallbladder [1,18]. In addition, some cases may arise in the AH or R-A sinus in the gallbladder walls [189,191,192,193]. Rowan et al. analyzed 19 cases of mural papillary mucinous lesions that arise in adenomyomatous nodules and form localized ICPNs [191]. All were characterized by a compact multilocular, demarcated, cystic lesion and many papillary lesions covered by mucinous epithelia. Their architecture, distribution and location were suggestive of the development of an underlying AH.

b. Pathogenesis

(1) Background lesions

In the conglomerated type, the microscopic intraepithelial neoplastic epithelial cells resembling high-grade BilINs with a micropapillary pattern are frequently found in the surrounding mucosa of the main polypoid neoplasms, and both lesions are continuous and shared many features [18,71,176]. Intraglandular involvement by similar neoplastic intraepithelial cells is also frequent as seen in high-grade BilINs, suggesting that high-grade BilINs with a micropapillary pattern may grow and protrude in the lumen, eventually resulting in grossly visible, conglomerated ICPN [71,176]. This type may reflect the “field-effect/defect” phenomenon [88,89]. Incipient ICPN may be a transitional lesion between high-grade BilINs and conglomerated ICPN. Recently, Nagao et al. [105] reported a mouse model that induced biliary neoplasm recapitulating BilINs and ICPN, thus supporting the above-mentioned suggestion.

However, the solitary type (ICPN without lateral and glandular intraepithelial neoplastic involvement or dysplasia) may have a different pathogenesis [71,176].

(2) Molecular and genetic changes

Akita et al. [187] identified somatic mutations in *STK11*, *CTNNB1* or *APC* in 71% of ICPN cases, and they were regarded as major driver mutations in ICPNs, while those alterations were exceptional in GBCs. ICPNs more commonly showed cytoplasmic and/or nuclear expressions of β-catenin than GBCs, indicating the participation of the *WNT* signaling pathway in the development of ICPN. While the KAS mutation is common in ICPN [194], mutations in *PT53* and *GNAS* at codon 201 are rare in ICPN [9,195].

Recently, Nagao et al. [105] developed a mouse model and reported that concurrent activation of the *KRAS* and *WNT* pathways induced biliary neoplasms that simultaneously resembled ICPN and BilIN, and that these precursors infrequently progressed to invasive carcinoma. In this model, genes associated with the *C-MYC* and *TGFβ* pathways were increased. They also reported using human ICPN cases in which both p-ERK and nuclear β-catenin were immunohistochemically expressed in more than a half of the ICPN cases of the gastric subtype, suggesting that the *KRAS* and *WNT* signaling pathways and the *C-MYC* and *TGFβ* pathways were also activated in human ICPNs. ALDH1A1, C-MYC, and p-SMAD3 were expressed in ICPN, suggesting that ICPN could be induced by biliary specific concurrent activation of the *KRAS* and *WNT* pathways. High-grade BilIN and ICPN form multifocal and extensive proliferations on the gallbladder mucosa, thus displaying the “field-effect/defect” phenomenon [88] and have a high propensity for invasion.

#### 3.3.7. Staging (TNM) and Prognosis

(1) Staging

ICPNs with HGD were staged as pTis. The staging of ICPNs with associated invasive carcinoma was similar to that of gallbladder carcinoma [42,43].

(2) Prognosis

Regarding post-operative OS, ICPN showed a favorable post-operative survival [18,176]. Type 1 is associated with more favorable post-operative outcomes than type 2, which has also been observed for IPNBs [176]. Berger et al. [196] found that biliary subtype adenocarcinomas arising from ICPNs had less aggressive pathologic features and showed improved survival compared to biliary subtype adenocarcinomas that did not arise from ICPNs. The three-year actual survival rate was 90% for cases without invasion and 60% for cases with invasion [18]. Invasion of ICPN has been reported to be related to associated with an unfavorable prognosis [18,176]. Patients with ICPN with stromal invasion were older than those without invasion [176].

#### 3.3.8. Related Diseases and Differential Diagnosis

##### Related Disease

Incipient ICPN: Preinvasive intraepithelial neoplastic lesions smaller than 1 cm that do not fulfill the size criteria of ICPN are occasionally encountered in the gallbladder without invasive carcinoma and without ICPN in other parts [18,175,176]. They present with one of the four subtypes and usually show LGD or HGD with a low-grade component. While they are smaller, they are similar to ICPN and could be an incipient ICPN [18], although they appear to be high-grade BilIN with micropapillary components greater than 3 mm. Mochizome et al. reported a relatively exophytic papillary pattern that does not meet the ICPN criteria and is referred to as “papillary neoplasia” [175].

Pyloric gland adenoma (PGA): PGA is defined as a localized and pedunculated polypoid lesion composed of packed pyloric glands showing a lack of association with surrounding intraepithelial neoplastic spreading or dysplasia [1,196]. Adsay et al. [18] previously proposed that ICPN was originally introduced as an umbrella term (intracholecystic papillary-tubular neoplasm [18]) to address all gallbladder tumoral intraepithelial neoplasms; therefore, ICPN was a heterogeneous, neoplastic polypoid lesion of the gallbladder [18]. In this context, PGA could be a solitary gastric subtype of ICPN (gICPN) without dysplastic surrounding mucosa and pyloric gland predominant type [18]. In the current WHO tumor classification [1], however, ICPN is separately described from PGA, although detailed discussion on the difference between PGA and gICPN, is lacking and the reasons why PGA was separated from ICONs has not been described. Reportedly, nuclear β-catenin accumulation has been immunohistochemically found in many cases of PGA [18,176], however such nuclear β-catenin accumulation was not found in gICPNs, including the solitary or conglomerated type, suggesting that PGA may undergo different molecular pathways than ICPN, including gastric subtype. In addition, cholesterol-containing foamy cells and squamoid morules were frequently observed in PGA, suggesting that PGA and gICPN can be differentiated histologically from each other [18,197,198]. PGA also presents with LGD, HGD, and an association with invasive carcinoma, such as in ICPN [1,185].

##### Differential Diagnosis

Adenomyomatous hyperplasia (AH): AH is a common non-neoplastic lesion in the gallbladder, and it is considered a tumor-like inflammatory lesion characterized by numerous branching R-A sinuses and extension of the subserosal area [79]. Some presenting a localized form could be confused with ICPN [79]. Multiple, prominent biliary epithelial proliferations with tubular-papillary architecture and minimal nuclear atypia are observed in association with chronic inflammation, fibroblastic proliferation and smooth muscle proliferation. ICPNs rarely arise in AH [192].

Papillary hyperplasia: Papillary hyperplasia shows crowded and tall columnar mucosal folds and papillary structures lined by columnar cells consistent with the biliary phenotype and present a focal or diffuse distribution [79]. This hyperplasia develops in pancreatobiliary malunion, metachromatic leukodystrophy, or cholecystitis. Dysplastic changes are absent, thus differentiating from ICPN [79].

Polypoid invasive carcinoma (PICA): In PICA, intraluminal polypoid components and invasive parts of the wall are composed of invasive adenocarcinoma, and few or no adenomatous or LGD components are found in intraluminal polypoid components [180,199]. In contrast, invasive ICPNs contain preinvasive adenomatous or low grade components in intraluminal mass-forming lesions; therefore, they are differentiated from PICA. However, the differentiation of ICPN of type 2 or HGD with considerable invasion from PICA remains controversial.

Metastatic carcinoma.

## 4. Conclusions

BTCs and precursors of the biliary tract were reviewed here as a member of the pancreatobiliary system in terms of the anatomy, embryology, pathologic features, and immunophenotypes including four cell lineages. A diagram to depicting the pathways for development of precursors and TBC is shown in Figure 4. Reports have shown that high-grade BilIN, IPNB, and ICPN are the major precursors of BTC and show characteristic features depending on the four subtypes and grading. Moreover, these precursors show intraepithelial lateral spreading to the surrounding mucosa and involve the peribiliary glands and the R-A sinus in the walls of the biliary tract. They are associated with invasive carcinoma at the time of surgical re0section, and their components are frequently found in the mucosa surrounding BTC. They may arise from the mucosa of the biliary tract, as well as and also the peribiliary glands and R-A sinus located in the walls of the biliary tract. Molecular and genetic features have now been clarified in terms of their counterparts in the pancreatobiliary system. A better clinical understanding of these precursors may lead to the early detection and development of targeted treatment strategies for BTC.

## Figures and Tables

**Figure 1 cancers-14-05358-f001:**
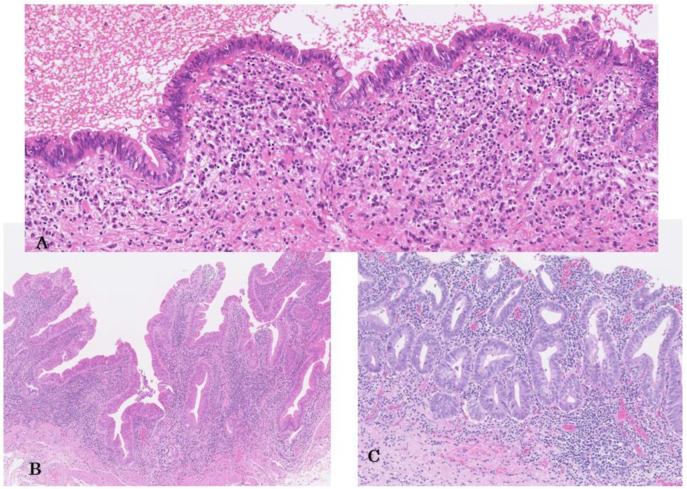
High-grade biliary intraepithelial neoplasm (high-grade BilIN); (**A**) Flat type. Lining epithelia of the extrahepatic bile duct show pseudostratified and hyperchromatic and swollen nuclei. Nuclear polarity is variably lost. ×300. HE staining. (**B**) Micropapillary type. Micro- and short- papillary structures are lined by neoplastic atypical epithelia. Gallbladder. ×80. HE staining. (**C**) Propria forming type. Atypical glands with mucosa propria formation, appearing as so-called intramucosal carcinoma of the stomach. Extrahepatic bile duct. ×100. HE staining.

**Figure 2 cancers-14-05358-f002:**
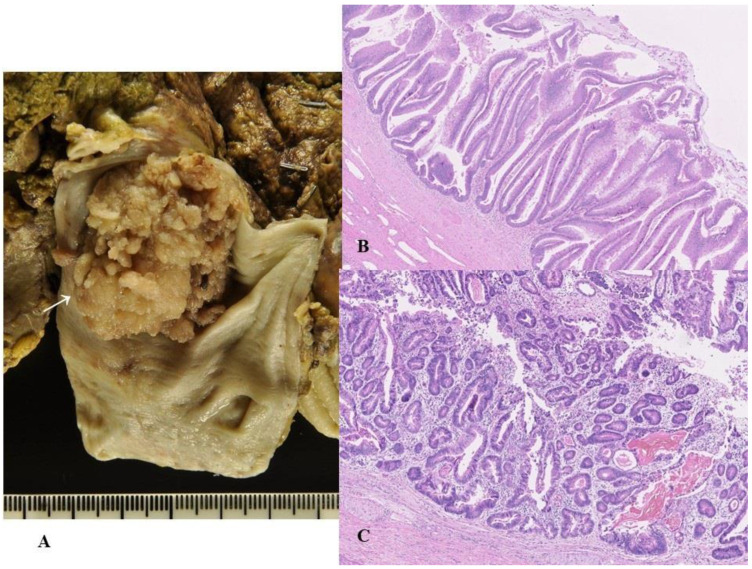
Intraductal papillary neoplasm of bile duct (IPNB) (**A**) Gross features. Papillary neoplasm in the dilated extrahepatic bile duct (arrow). (**B**) Villous neoplasm with fine fibrovascular stalk and low-grade dysplastic changes. Type 1 IPNB of intestinal subtype (villous/papillary predominant type). Intrahepatic large bile duct. ×100. HE staining. (**C**) Tubular neoplasm with irregular structures and high-grade dysplastic changes. Type 2 IPNB of intestinal subtype (tubular predominant type). ×100. HE staining.

**Figure 3 cancers-14-05358-f003:**
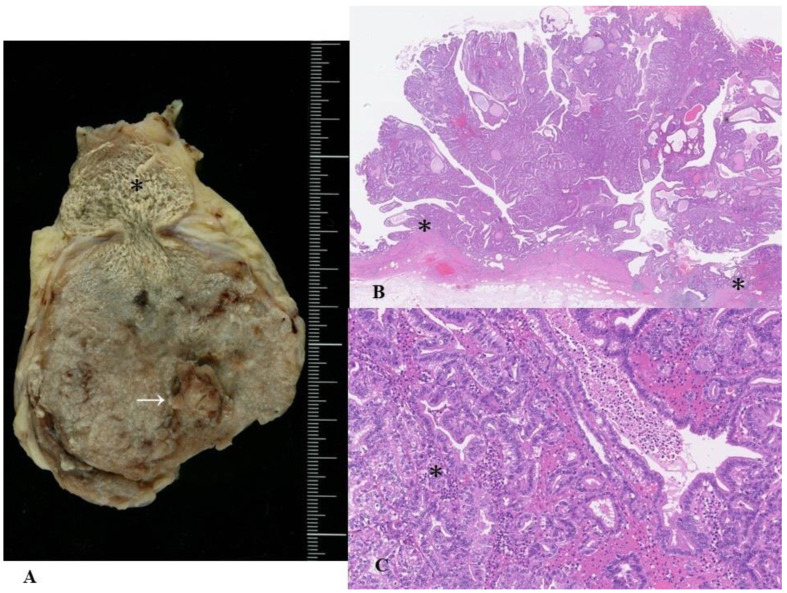
Intracholecystic papillary neoplasm of gallbladder. (**A**) Gross features. Polypoid or mass-forming lesion is seen in the fundus. In the surrounding mucosa showing rough appearance, small papillary or coarse granular lesions are multiply found. The mucosa of the neck is spared and shows cholesterosis (*). Conglomerated type. ×40. HE staining. (**B**) Conglom-erated papillary or polypoid neoplastic lesions with shorter papillary and flat neoplastic changes in the peripheral area (*). HE staining. (**C**)High power magnification shows high-grade gastric differentiation, and some parts show low-grade differentiation or adenomatous features (*). High-grade dysplastic ICPN of gastric subtype. ×90. HE staining.

**Figure 4 cancers-14-05358-f004:**
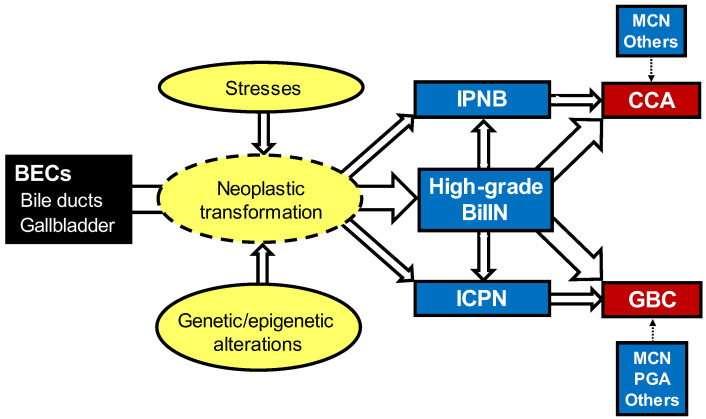
Biliary epithelial cells (BECs) of mucosal lining epithelia and of peribiliary glands (bile ducts) and R-A sinus (gallbladder) undergo neoplastic transformation under environmental stresses and with genetic/epigenetic alterations. As a result, precursor lesions such as intraductal papillary neoplasm (IPNB), high-grade biliary intraepithelial neoplasm (BilIN) and intracholecystic papillary neoplasm (ICPN) develop, and these precursors progress to biliary tract carcinoma: cholangiocarcinoma (CCA) and gallbladder carcinoma (GBC). High-grade BilIN may also relate to the development of IPNB and ICPN. Other precursors such as mucinous cystic neoplasm (MCN) and pyloric gland adenoma (PGA) play a minor role in the development of CCA and GBC.

**Table 1 cancers-14-05358-t001:** Characteristic features of four subtypes in pancreatobiliary neoplasms based on cell lineages.

	Intestinal Subtype	Gastric Subtype	PB Subtype	Oncocytic Subtype
Histologic features	Columnar epithelia with single-layered or pseudostratified cigar-shaped nuclei and with basophilic, amphophilic cytoplasm with apical vesicular mucin. Tubular, papillary/villous and serrated structures. Some show villous/papillary pattern, while the other showed tubular or tubule-villous pattern.	Regional growth of papillary and tubular (glandular) neoplastic epithelia resembling gastric foveolar epithelia and pyloric glands. Proportion of these two components is variable: some cases showed predominantly foveolar pattern or pyloric gland pattern, while the other showed equal amount of both components	Single-layered, small to medium cuboidal/low-columnar, neoplastic- epithelia. Centrally or basally located, nuclei and slightly acidophilic cytoplasm. Epithelia cell and nuclei resembling simple epithelia of bile or pancreatic duct. Many fine papillary structures with many ramifying branches of thin fibrovascular stalks.	Single- to multi-layered medium-sized cuboidal to low-columnar epithelia with eosinophilic. granular cytoplasm. Frequent secondary lumina formation. Arborized papillary and/or cribriform formations. Thin or edematous or inflammatory stroma
Immunohisto- chemical features	Frequently positive for CDX2 and CK20. Goblet cells positive for MUC2.	Foveolar components positive for MUC5AC, and pyloric glands positive for MUC6.	Positive for CK7, S100P and MUC1.	Positive for gastric-type mucin (MUC5AC and MUC6) and for HepPar-1

**Table 2 cancers-14-05358-t002:** Precursor lesions of pancreatobiliary system.

Anatomical Locations	Precursors
Intrahepatic large bile duct Perihilar bile duct Distal bile duct	BilIN (low-grade and high-grade) IPNB MCN
Gallbladder	BilIN (low-grade and high-grade)
	ICPN
	MCN
	Pyloric gland adenoma
Pancreas	PanIN (low-grade and high-grade)
	IPMN
	MCN
	IOPN
	ITPN
Ampulla	Intestinal adenoma (low-grade and high-grade)
	IAPN
	Flat intraepithelial neoplasm

No established precursors are reported in the intrahepatic small bile duct; BilIN, biliary intraepithelial neoplasm; IPNB, intraductal papillary neoplasm of bile duct; MCN, mucinous cystic neoplasm; ICPN, intracholecystic papillary neoplasm of gallbladder; IPMN, intraductal papillary mucinous neoplasm of pancreas; IOPN, intraductal oncocytic papillary neoplasm of pancreas; ITPN, intraductal tubular papillary neoplasm of pancreas; IAPN, intra-ampullary papillo-tubular neoplasm.

**Table 3 cancers-14-05358-t003:** Differences between low-grade and high-grade BilIN (biliary intraepithelial neoplasm).

	Low-Grade	High-Grade
Histology	Flat/pseudopapillary/micropapillary	Flat/pseudopapillary/micropapillary/thickened
	appearance	mucosa appearance
	Hyperchromatic nuclei	Hyperchromatic and irregular nuclei
	Increased N/C ratio	Pleomorphic, bizarre nuclei, increased N/C ratio
	Nuclear stratification	Complex nuclear stratification
	Preserved nuclear polarity	Loss of nuclear polarity
Biliary mucosa involvement	Relatively small foci or areas	Relatively extensive area
Involvement of peribiliary glands/RAS	Infrequent	Frequent
Ki-67 index	Mildly to moderately increased	Markedly increased
Immunostainings		
S100P	Mild to moderately increased	Diffuse and strongly positive
P53	Usually negative	Frequently positive
P16	Relatively preserved	Decreased

**Table 4 cancers-14-05358-t004:** Intraepithelial lateral spreading and glandular involvement by neoplastic epithelia in the bile duct and gallbladder in high-grade BilIN (biliary intraepithelial neoplasm), IPNB (intraductal papillary neoplasm of bile duct) and ICPN (intracholecystic papillary neoplasm).

	Intraepithelial Lateral (Superficial) Spreading around High-Grade BilIN, IPNB and ICPN	Glandular Involvement Beneath High-Grade BilIN, IPNB and ICPN
Affected tissues	Surrounding mucosa (non-neoplastic lining epithelia and mucosal glands) of the bile duct and gallbladder	Bile ducts: Peribiliary glands and their conduits in the duct wall and periductal tissue
		Gallbladder: RAS and non-neoplastic tubules in the wall and subserosa
Characteristic growth patterns	Neoplastic epithelia show abrupt transition against non-neoplastic epithelia	Mixture of neoplastic epithelia and non-neoplastic epithelia within peribiliary glands and their conduits or RAS and non-neoplastic tubules
	Neoplastic epithelia show replacing spread in the mucosa for considerable areas and length (occasionally involving more than one anatomical segments of biliary tract)	Continuous and discontinuous involvement of peribiliary glands and their conduits or tubules and RAS from neoplastic epithelia of luminal surface
	Spreading neoplastic epithelia show the same cell lineage of the high-grade BilIN, IPNB or ICPN	Involving neoplastic epithelia show the same cell lineage of the high-grade BilIN, IPNB or ICPN
	Not associated with desmoplastic reaction	Not associated with desmoplastic reaction
Significance	Growth and extension of high-grade BilIN and IPNB and ICPN	Growth and extension of high-grade BilIN and IPNB and ICPN
	Surgical margin at operation	Diagnostic pitfall for invasive growth

**Table 5 cancers-14-05358-t005:** Characteristic features of modified two-tiered grading (type 1 and type 2 subclassification) of IPNB (intraductal papillary neoplasm of bile duct).

	Type 1	Type 2
Findings for classification		
Prototypic structures of four subtypes	Retained and regular	Variably lost and irregular
Gastric, pancreatobiliary subtypes	Similar to prototypes of IPMN * of pancreas	Variably different from prototypes of IPMN * of pancreas
Oncocytic subtype	Similar to IOPN ** of pancreas	Variably different from prototype of IOPN ** of pancreas
Intestinal subtype	Similar to intestinal adenoma of ampulla	Variable different from prototype of intestinal adenoma of ampulla
Cellular and nuclear atypia	Low-grade or low- and high-grade	High-grade including overt malignancy
Cribriform and solid growth	None	Not infrequent
Necrosis	None	Not infrequent
Other features		
Location intrahepatic	58–68%	14–27%
extrahepatic	32–35%	48–64%
both	7%	22%
Mucus hypersecretion	29–86%	12–21%
Invasion	6–50%	60–94%

* intraductal papillary mucinous neoplasm; ** intraductal oncocytic papillary neoplasm.

**Table 6 cancers-14-05358-t006:** Main differences of solitary vs. conglomerated ICPN (intracholelcystic papillary neoplasm).

	Solitary Type	Conglomerated Type
Gross features	Pedunculated, polypoid	Sessile, multiple or coalescent, excrescences
Border	Well demarcated from the surrounding mucosa	Continuous with surrounding granular or rough mucosa
Involvement	One to more areas *	Localized to one area
Background mucosa lesion	No or very limited intraepithelial neoplasm, dysplasia	Wide spreading of micropapillary and flat intraepithelial neoplasm, dysplasia
Stromal Invasion	Infrequent	Frequent
Incidence	About 25% of ICPN	About 75% of ICPN

* Areas: neck, body and fundus.

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
