# Peer review of "Pathologies of Precursor Lesions of Biliary Tract Carcinoma"

_cancers, 2022, doi:10.3390/cancers14215358_

Round 1

Reviewer 1 Report

This is a comprehensive review of pathologies of precursor lesions of the biliary tract carcinoma (BTC). The authors described histologies, cell lineages, stages, prognosis and related diseases of precursors of BTC. My minor comments are as followings. 

(1) I would suggest adding reviews on the genetics of BTC precursor and BTC, such as genetic etiology, somatic mutational profiles of precursors and BTC tumors.

(2) Including epidemiology data of precursors and BTC would be great. 

(3) Diagnostic biomarkers, prevention, and treatment of precursor and BTC should be described. 

Author Response

Responses to Reviewer 1:

  • I would suggest adding reviews on the genetics of BTC precursor and BTC, such as genetic etiology, somatic mutational profiles of precursors and BTC tumors.

Thank you for nice comment. We already described and discussed the genetics of BTC precursors in individual lesions, separately and considerably. So, we added the genetics of TBC in the section of BTC, and cited newly 7 references (#53-59), as follows, in the revised mansuscript.

Page 7, 4th and 5th para:  2.2.5.  Genetic alterations in LD-iCCA, pCCA, dCCA, and GBC

More than 50% of GBCs and approximately 50% of extrahepatic CCAs harbor alterations in TP53 alteration [53,54]. Common mutations in GBC include CDKN2A or CDKN2B (19%), ARID1A (13%), PIK3CA (10%) and CTNNB1 (19%), and amplifications of ERBB2 have been also reported [55-57]. Specific extrahepatic CCA alterations include PRKACA/PRKACB fusion, ELF3 mutation and ARID1B mutation [58]. KRAS mutations were observed in 20-30% of extrahepatic CCAs, while an increase in KRAS mutations are closely related to GBCs with pancreatobiliary malunion [59].

Genetic alterations in the precursors of BTCs are discussed in individual lesions in the following sections.

  • Including epidemiological data of precursors and BTC would be great.

Thank you for nice suggestion. We added the epidemiological data and cited newly 3 references (#44,#45,#46), in the revised manuscript, as follows.

Page 5, 4th para;

 2.2.1. Epidemiology

The incidence of GBC varies geographically and ethnically. The incidence of GBC is the highest in Chilie with anincidence among females of 27.3 cases per 100.000 person-year [1,44]. The incidence is also higher in Indica, eastern Asia, and central and eastern European countries [1,45]. GBC occurs predominantly in females with gallstones in Chilie, while the association with gallstones in eastern Asia is weak [1,44,45]. The incidence of extrahepatic CCAs ranges from 0.53 to 2 cases per 100,000 person-year worldwide, with higher frequency in east Asian countries [45a]. In some regions of east-Asia, particularly the Republic of Korea and Thailand, liver fluke infestations and hepatolithiasis are endemic [46].

Page 10, 2nd  para; 3.1.1.3. Epidemiology

Low-grade BIlIN is seen in as many as 15% of gallbladders with lithiasis, and high-grade BilIN in 1-3.5%. in regions where GBC is endemic [78-79]; incidences were reported in <5% and < 0.1% in North America [80]. The epidemiology of low-grade and high-grade BilIN in the bile ducts has not been reported because they are seldom biopsied. However, high-grade BilIN lesions are frequently found in the surrounding mucosa of BTC [22,70,81,82], although the epidemiology of such BilINs has not been surveyed epidemiologically.

Page 16, 3rd  para:  

2.1.1. Clinical features and background and epidemiology

IPNB is typically diagnosed in middle-aged or elderly adults and has a slightly higher occurrence in males [7,8,114-117]. IPNB occurs worldwide, but its incidence varies geographically [7,8,118]. IPNBs account for 10-38% of all bile duct tumors in eastern Asia but only 7-12% of all bile duct tumors in North American and European countries., and a higher incidence is noted in South-East and Far-East Asian countries [7,8,119,120]. Risk factors include hepatolithiasis, liver fluke infections, primary sclerosing cholangitis (PSC), congenital biliary tract diseases and exposure to chemicals such as chlorinated organic solvents [7,8,121,122]. These usually occur as single tumor and/or occasionally as multiple tumors, and can present clinically as large duct obstructions with recurrent abdominal pain, cholangitis and cholestatic hepatic dysfunction [7,8,115,123].

Page 28- 2nd para

3.3.1. Clinicopathological features and background and epidemiology

ICPNs are twice as common in females as in males, and the patients range in age from 20 to 94 years (mean: 61 years) [1,18]. Almost half of all patients with ICPN present with right upper outer quadrant pain. Almost half of these cases are radiologically diagnosed as gallbladder cancer [18]. Approximately 6% of gallbladder carcinomas that arise in association with ICPN [1,18,175,177]. Approximately a half of ICPN cases are associated with invasive carcinoma (ICPN associated with an invasive carcinoma) at the time of surgical resection [18,175]. ICPN is found in 0.4% to 1.5% of all cholecystectomies, and its incidence ranges from a rare disease to 23.5%-28% of primary epithelial gallbladder neoplasms [18,175,177], though there have been no systematic geographical reports on its incidence of ICPN. There are no known etiologic factors in ICPN [1], although several biliary diseases such as pancreatobiliary malunion, xanthogranulomatous cholecystitis, lymphocytic cholecystitis, and segmental adenomyomatosis are associated with ICPN. Cholecystolithiasis is found in 20% of the ICPN cases

  • Diagnostic biomarkers, prevention, and treatment of precursor and BTC should be described.

Thank you for your nice comments. In this review paper, we aim to review the pathologic aspects of the precursors of the biliary tract carcinoma. So, we think that the detailed description of diagnostic markers, prevention, and treatment of precursors and BTS are beyond the scope of this review paper. So, we added briefly these topics as follows.]

Page 8, 3rd  para;

2.4.Treatments and diagnostic approaches of BTCs and precursors of the biliary tract

The treatments of BTC is dependent on its anatomical location and stages based on the TNM classification [42,43]. Biliary tract precursors, particularly IPNBs, are not infrequently associated with invasive carcinoma [7,8,38], thus, they are recommended to be treated similarly to BTCs [7,8]. Recent progress in biomarkers and diagnostic and therapeutic approaches for biliary neoplasms, including BTCs, has been reported in the IASGO textbook [63].

Reviewer 2 Report

Reviewer comments:

Comments to the Author

The article on “Pathologies of precursor lesions of the biliary tract carcinoma” by Dr. Nakanuma is emphasizing on biliary tract carcinoma (BTCs) and precursors of the biliary tract that belong to pancreatobiliary neoplasms. Authors have well discussed about the recent progress on the pathologic features of the precursors of the biliary tract, particularly high-grade BilIN, IPNB and ICPN, and BTCs and pancreatic and ampullary counterparts are also referenced.

This article has made an impressive effect in terms of elaborating and bringing the details on common precursors of the biliary tract.

They not only tried to compile relevant studies focused on the current updates but also presented very well with substantial discussion of results and postulated according to the evidence provided. The review organization is impressive, and the tables and figures provided were comprehensive. The references are appropriate and timely.

Minor criticisms

• Please undergo a thorough check of the manuscript for typographical and grammatical errors.

Author Response

Responses to Reviewer 2:

It would have been more interesting if they would have included more pictorial diagrams to depict the various pathways or mechanisms involved.

Thank you for nice comments. We newly added newly Figure 4 depciting the developmental pathway of precursors and BTC. Accordingly, the following sentences were added in the revised manuscript.

Page 36, line 1-2

of the anatomy, embryology, pathologic features, and immunophenotypes including four cell lineages. A diagram to depicting the pathways for development of precursors and TBC is shown in Fig.4. Reports have shown that

Figure legend, page 58-59, Figure 4.

Figure 4. Biliary epithelial cells (BECs) of mucosal lining epithelia and of peribiliary glands (bile ducts) and R-A sinus (gallbladder) undergo neoplastic transformation under environmental stresses and with genetic/epigenetic alterations. As a result, precursor lesions such as intraductal papillary neoplasm (IPNB), high-grade biliary intraepithelial neoplasm (BilIN) and intracholecystic papillary neoplasm (ICPN) develop, and these precursors progress to biliary tract carcinoma: cholangiocarcinoma (CCA) and gallbladder carcinoma (GBC). High-grade BilIN may also relate to the development of IPNB and ICPN. Other precursors such as mucinous cystic neoplasm (MCN) and pyloric gland adenoma (PGA) play a minor role in the development of CCA and GBC.

Figure 4 was newly added.

Reviewer 3 Report

Nakanuma et.al. in their review identified BilIN, IPNB and ICPN as the major precursors of the BTCs based on their anatomy, embryology, pathologic features, and immunophenotypes. In their review, they emphasized on the importance of peribiliary glands, R-A sinus in the walls of the biliary tract and the intraepithelial lateral spreading to surrounding mucosa. The strongest part of this review is that all the aspects of the precursors including their anatomical locations, clinical features, backgrounds, gross and microscopic features etc. have been discussed in details. It would have been more interesting if they would have included more pictorial diagrams to depict the various pathways or mechanisms involved. The present review has a great clinical importance and would be beneficial to researchers especially in the field of BTCs. Hence, I strongly recommend it for publication.

Author Response

Responses to Reviewer 3:

Thank you for your nice and thoughtful comments. I think this reviewer did not ask revisions, so no revisions were made in the revised manuscript.